# Characterization of diffusing sub-10 nm nano-objects using single anti-resonant element optical fibers

Torsten Wieduwilt [1,4], Ronny Förster[1,4], Mona Nissen[1,2], Jens Kobelke[1] & Markus A. Schmidt [1,2,3] ✉

Accurate characterization of diffusing nanoscale species is increasingly important for revealing processes at the nanoscale, with fiber-assisted nanoparticle-tracking-analysis representing a new and promising approach in this field. In this work, we uncover the potential of this approach for the characterization of very small nanoparticles (<20 nm) through experimental studies, statistical analysis and the employment of a sophisticated fiber and chip design. The central results is the characterization of diffusing nanoparticles as small as 9 nm with record-high precision, corresponding to the smallest diameter yet determined for an individual nanoparticle with nanoparticle-tracking-analysis using elastic light scattering alone. Here, the detectable scattering cross-section is limited only by the background scattering of the ultrapure water, thus reaching the fundamental limit of Nanoparticle-Tracking-Analysis in general. The obtained results outperform other realizations and allow access to previously difficult to address application fields such as understanding nanoparticle growth or control of pharmaceuticals.

The precise analysis of the size and concentration of diffusing nanoscale species is increasingly important to understand, in particular, interactions and processes at the nanoscale level. Examples of applications include understanding the in situ growth of nanoparticles (NPs)[1], monitoring of water contaminants[2], or quality control of medical products (e.g., vaccinations[3]). Here, many relevant species have sizes in the range <50 nm (e.g., viruses, quantum dots[4,5]), placing very high demands on analytics.

In principle, optical approaches are particularly well suited to decipher such nanoscale processes due to their non-invasive operational principle. However, a direct microscopic characterization of NPs is unfeasible if the diameters of the NPs falls below the diffraction limit. One approach applied in this context is dynamic light scattering (DLS), evaluating the diffusive properties of ensembles of NPs. Here, even though nano-objects such as proteins or molecules can be characterized[6], polydisperse samples are difficult to handle, very high and partially inappropriate specimen concentrations are required, and larger NPs are intrinsically preferred[6,7]. Moreover, DLS does not allow analyzing the dynamics of NPs at the level of individual objects, since the properties of an ensemble of NPs and not those of individual particles are determined.

A conceptually different approach that operates at the single species level is nanoparticle-tracking analysis (NTA). This technology statistically evaluates individual NP trajectories, making it possible to characterize small particle concentrations[8–10] and polydisperse samples, both in contrast to DLS[9,11,12]. Examples of successful NTA applications include the characterization of viruses, extracellular vesicles or nanomotors[8,13,14], or pharmaceutical quality control[3]. Note that crucial to NTA is the correlation between achievable accuracy of determined diameter (coefficient of variation (CV)) and trajectory length (i.e., no.

[1]Leibniz Institute of Photonic Technology, Albert-Einstein-Str. 9, 07745 Jena, Germany. [2]Abbe Center of Photonics and Faculty of Physics, Friedrich Schiller University Jena, Max-Wien-Platz 1, 07743 Jena, Germany. [3]Otto Schott Institute of Material Research, Friedrich Schiller University Jena, Fraunhoferstr. 6, 07743 Jena, Germany. [4]These authors contributed equally: Torsten Wieduwilt, Ronny Förster. ✉e-mail: markus.schmidt@leibniz-ipht.de

of frames per trajectory)[15], suggesting to use of approaches that enable the acquisition of as-long-as-possible trajectories.

Since fluorescence-based methods require molecular markers, which may alter the diffusive properties of, in particular, very small NPs[16], elastically scattered light has proven to be advantageous for NTA. Here, the remaining background scattering limits the diameter that can be determined and thus the precision, making the detection of very small NPs a major challenge[17]. Here, the gold standard instrument in this context is the Nanosight NS300 (Malvern), which can detect gold NPs down to 10 nm and measure them with a CV of 33% (CV = $\sigma/\mu$). This rather high CV results from the NPs leaving the focus and/or field of view (FoV), thus limiting achievable trajectory lengths[15].

A novel NTA approach introduced by the authors is based on tracking NPs in microstructured optical fibers, defining the concept of fiber-assisted NTA (FaNTA). The key feature of FaNTA is the confinement of NPs in fiber-integrated microchannels. It leads to high-intensity pseudo light line illumination through the optical mode and fast readout times. Particularly relevant is the confinement of the NPs to the light-guiding sections, preventing NPs from leaving the illuminated volume or the focal plane of the microscope. This results in exceptionally long observation times (i.e., trajectories with a very large number of frames) of rapidly diffusing objects, leading to high statistical significance in the diameter determination of individual NPs, even for very small particles[18,19]. Note that in other NTA implementations, precision is defined with respect to the entire NP ensemble (mainly resulting from the low number of frames), while FaNTA gives a precision per individual NP. In addition to solid-core fibers[20,21], hollow-core fibers (HCFs) represent a very promising FaNTA platform. Here, the central core serves simultaneously as a fluidic micro- and light-guiding channel, maximizing illumination performance and ensuring real-time analyte exchange. As an example, the authors were able to characterize ensembles of gold NPs and bacteriophages[22]. A recent fiber design that includes one antiresonant-element (ARE) showed significantly improved imaging properties, allowing for the characterization of mixtures of NPs that are indistinguishable in the case of DLS[23]. These achievements clearly suggest that FaNTA has the potential to characterize NPs of extremely small size at unprecedented levels of precision, forming the motivation for this work.

In this work, we unlock the potential of FaNTA for the characterization of very small NPs (<20 nm) by means of experimental investigations and detailed statistical analysis (Fig. 1a). The study is based on an optimized experimental configuration and a sophisticated fiber design that consists of one single antiresonant channel, both having tailored properties to achieve optimized FaNTA performance. Based on the results of our previous work that concentrated on the microscopic imaging through the ARE and the capabilities to separate bimodal mixtures[23], we focus here on uncovering the potential of the ARE-concept with respect to the characterization of NPs with extremely small diameters, where we were able to uncover the fundamental limits and influences. In addition to the aspects addressed in[23] we discuss in this work issues such as the impact of the microchannel confinement on diffusion and data analysis, the realization of a novel chip design, or the influence of photon pressure, and present an experimental reliability study of the ARE-concept. All the issues addressed in this study are related to the characterization of very small NPs. The key result is the characterization of diffusing NPs as small as 9 nm with record low CV values (CV = 13%, an example of tracking such small NPs with FaNTA is shown in Fig. 1b−d). The detectable scattering cross-section is here only limited by the background scattering of the ultrapure water, thus reaching the fundamental limit of this approach.

## Results

### Design of single antiresonant element fiber

Conceptually, FaNTA is based on microscopically tracking the motion of diffusing NPs inside the microchannels of microstructured optical fibers. The key to the FaNTA realization discussed in this work is a fiber with a single ARE (single element fiber, SEF), providing effective single-mode guidance directly inside the water-filled ARE (Fig. 2d, details can be found in ref. 23). The main advantage of this arrangement is the single connection point of the ARE with the supporting capillary (Fig. 1a), yielding excellent tracking properties and nearly aberration-free imaging as the number of disturbing interfaces between NP and microscope objective is reduced to a minimum. As shown in ref. 23 by a cross-correlation analysis correlating the recorded image of a single NP with the ideal point spread function, the resulting image is diffraction-limited for almost all transverse positions of the NP. The wall thickness ($w = 730 \pm 20$ nm) of the ARE element (diameter $d_c = 17 \pm 0.5 \, \mu$m) was chosen to yield low modal losses (0.4 dB/ cm in simulation and experiment[24], see Supplementary Information Sec. SI 14 (Fig. S-12) for further details) at the operation wavelength ($\lambda = 532$ nm). More details on defining the spectral operating range can be found in this work[24]. From a waveguide and imaging perspective, the presented fiber design has been optimized with respect to several aspects, including the effect of the ARE-jacket junction, modal losses, or the size of the microchannel (for more details, see the Supplementary Information Secs. SI 12 and SI 13, Figs. S-10, 11). From the diffusion point-of-view, the comparably large channel diameter allows to avoid the influences of a changing viscosity near the wall and the saturation of the MSD values used for the fitting. More details and several example calculations on this can be found in the Supplementary Information (Secs. S-15 and S-16, Figs. S-13, 14). The fiber was produced in-house through the stack-and-draw procedure (scanning-electron-micrograph (SEM) images are shown in Fig. 2b, c) and the microchannel has an NA of 0.024 (water-filled).

### Optimization of FaNTA setup

To unlock the limits of the FaNTA realization discussed here, we imaged and analyzed 9, 20, 30, and 50 nm ultra-uniform gold and 50 nm latex particles (details of the NPs can be found in Table S1 of the Supplementary Information). Note that we deliberately selected NPs with defined and inserted surfaces from companies providing well-established NPs to avoid surface-related effects which may impact diffusion. The ideal specimen concentration is around 100 NP/nL, depending on the particles' diffusion coefficient and scattering cross-section. For comparison, DLS measurements using a commercial device have been performed at the recommended concentration of $10^5$ NP/nL (Zetasizer Nano ZS and Ultra; Malvern Panalytical), which is substantially higher than the one used in the NTA experiments.

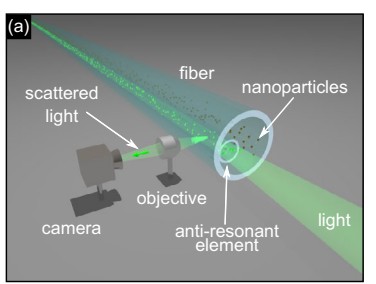
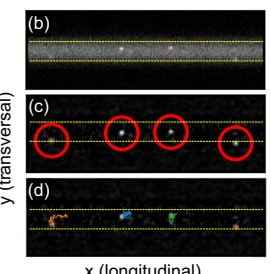

**Fig. 1 | The concept of single antiresonant-element (ARE) fiber-assisted nanoparticle-tracking analysis (FaNTA) applied to track sub-10-nm nano-objects. a** Sketch of the methodology. An example of tracking such small nanoparticles with FaNTA can be seen in the images to the right. **b** Example of a selected frame showing 9 nm gold NPs diffusing inside the antiresonant element. **c** Processed image showing the localization of the NPs (red circles). **d** Corresponding measured trajectory of several tracked nanoparticles. In all images on the right-handed side, the horizontal yellow dashed lines indicate the wall of the ARE.

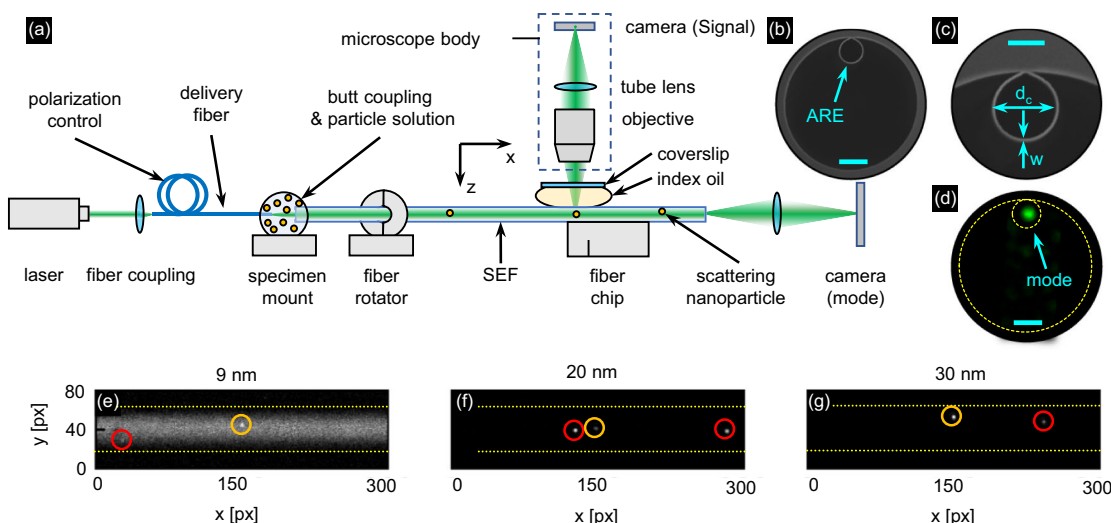

**Fig. 2 | Optimized experimental setup used in the FaNTA experiments. a** Sketch of the setup with the relevant details. **b** SEM image of used ARE fiber (the scale bars shown in (**b**) and (**d**) refers to 20 μm, in (**c**) to 10 μm). **c** is a close-up view of the ARE. **d** Measured mode profile at the operation wavelength. The bottom row shows example raw images of tracked gold nanospheres (**e** 9 nm, **f** 20 nm, **g** 30 nm). Tracked particles are marked red. The orange circle shows a particle with typical SNR and brightness which increases with growing NP diameter. The yellow dashed line marks the channel wall.

The optimization of the experimental setup is essential for revealing the limit of the FaNTA approach and hence represents an essential result of this study. The setup is shown in Fig. 2a: visible cw-light (VERDI G18, $\lambda = 532$ nm, Coherent) is coupled into a single-mode delivery fiber (S405-XP, Thorlabs), which has a customized low NA fiber (mode field diameter 8.4 μm @ $\lambda = 532$ nm, NA = 0.05, IPHT) spliced to its end for a background optimized butt-coupling to the SEF sample. A drop of the NPs suspension is applied to the coupling point, so that the entire fiber fills by capillary action. The scattering signal of the NPs is transversely imaged by standard microscopy equipped with a 10x/0.25 Plan N objective (Olympus) and an CMOS camera (acA4096-40um, Basler). To optimize the butt-coupling efficiency (~90%) with respect to the launching efficiency of the fundamental mode, the output mode (example mode image shown in Fig. 2d) is imaged by another camera (DCC1240C-HQ, Thorlabs). More details on the optimization of the incoupling can be found in the Supplementary Information (Sec. SI 19, Fig. S-17). The guided laser power used at the position of the particles ranges from 0.2 mW for the 50 nm gold NP to 70 mW for the 9 nm gold NP, which was empirically determined.

One key to observe the diffusion of very small NPs is the optimization of the signal-to-noise ratio (SNR) of the scattered light, optimized here through the following actions: (i) As the scattered far-field pattern resembles a Hertzian dipole, the scattered power in the microscope is maximized by establishing linear input polarization perpendicular to the microscope axis. (ii) Aberrations which may arise due to the microstructure of the SEF are suppressed, if the SEF is rotated such that the ARE faces the objective[23]. (iii) Spherical aberrations and astigmatism originating from the curved SEF surface are avoided by embedding the fiber in an index-matching gel and covering it with a fused silica coverslip ($n(\lambda = 532$ nm, 20 °C) = 1.46) for all; more information in the supplementary information (SI 1). Together results in diffraction-limited imaging (Fig. 2d–f[23]).

The main sources of background signal are associated with iso-tropically scattered light from water, contamination, and modal leakage. The water contribution is minimized by a median filter, while those from contamination are avoided through mechanical filtering (Anotop syringe filter 0.02 μm, Whatman). In contrast to that, the background contribution resulting from modal leakage is emitted under a small and defined angle, which is well below the acceptance angle of the objective. This light can scatter on walls and interfaces towards the objective, resulting in a strong and fluctuating speckle background. A specially made chip prevents this light from entering the objective or the FoV (see SI 1). In conclusion, every NP is imaged by a diffraction-limited point spread function with no background except the unavoidable water trace[23].

Kindly note that an interaction of the NP with the wall of the channels could not be observed in any of the previous experiments using gold NPs (even in fibers with much smaller fluidic channels[18,19]). Thus, an influence of a surface interaction on the diameter determination can be excluded in the experiments discussed in the following.

### Characterization of nanoparticles and error analysis

The potential of our approach is clearly visible from the three example frames shown in Fig. 2e–g: NPs with very small physical diameters can be tracked by our approach, with Fig. 2e showing that NPs with a diameter below 10 nm are particularly challenging due to the presence of background signal, which we attribute to the scattering of the sur-rounding water (SI 3).

Within the context of NTA, the statistical error in the MSD analysis $\sigma_d$ is dominated by two practical issues: First, a successful tracking of indistinguishable NPs requires a minimum distance between the NPs, which cannot be achieved permanently due to their random move-ments. Thus, the time each NP can be tracked is limited, demanding to use of the highest possible frame rate to maximize trajectory length $N_f$ and precision (SI 2, Eq. (3)). Note, that at concentrations beyond those used here, NPs may cross each other many times and the tracking code may not be able to differentiate which NP belongs to which trajectory, resulting in shorter trajectories. Second, the SNR in each image has to be high enough for a successful localization of the NP and an opti-mized fitting of the MSD-curve. We successfully imaged the 9 nm gold NP at 450 fps (Fig. 2e and used this frame rate, for a fair comparison, of the remaining specimen. The retrieved diameter is independent of exposure time and frame rate (see SI 4).

A particularly important dependency is the correlation between the precision (i.e., CV) and the number of evaluated particles $N_p$. To exemplify this dependency, Fig. 3a–e show the result of the MSD analysis for a selected ensemble (20 nm gold) taking into account only NP with trajectory length above a certain threshold $N_{f,min}$ (indicated by the colored horizontal lines in Fig. 3a). It is evident from this example that the precision of the retrieved NP diameter ($\sigma_d$) strongly depends on the required trajectory length $N_{f,min}$ (e.g., $\sigma_d = 5.9$ nm @ $N_{f,min} = 50$ (Fig. 3e), $\sigma_d = 2.0$ nm @ $N_{f,min} = 1000$ (Fig. 3b)). Note that the curves

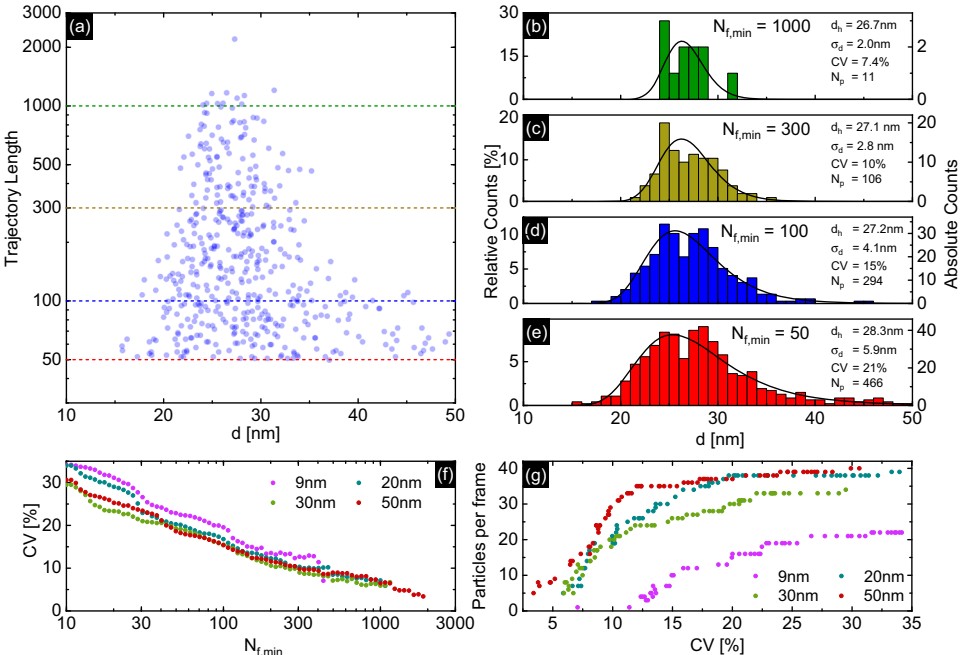

**Fig. 3 | Characterization of different nanoparticle ensembles using FaNTA.**
**a** Example of a measurement of an ensemble of 20 nm NP, showing trajectory length $N_f$ as a function of retrieved diameter. The differently colored lines indicate the considered minimum trajectory length $N_{f,min}$ used to calculate the histograms (probability of hydrodynamic diameter) shown in (**b**–**e**). **f** CV as a function of $N_{f,min}$. **g** Number of particles per frame as a function of CV. It is evident from the plots that low CV values require long trajectories, which, however, reduces the number of successfully evaluated particles in each frame.

presented in the histograms do not result from fitting, but rather from averaging of the data. In this process, the trajectories are weighted according to their length, so that longer trajectories, i.e., trajectories with higher accuracy, have a stronger contribution. This is where the great advantage of fibers becomes clear, as unlike other NTA approaches, FaNTA allows very long trajectories to be retrieved per NP through the channel-mediated confinement, resulting in statistics with very high significance (more details on the correlation between trajectory length and the precision of the ensemble statistics can be found in the Supplementary Information (Sec. SI 17, Fig. S-15)). Note that with increasing demand on precision, fewer trajectories have the necessary length, leading to more sparse evaluation (e.g., in Fig. 3g, only five NPs have sufficient trajectory length to contribute to the CV).

Applying the above analysis to all measured NP ensembles confirms that the CV improves with increasing $N_{f,min}$ (Fig. 3f), following a monotonic evolution (cf. SI 2 Eq. 3). Figure 3g shows that the smaller and thus faster the NPs are, the fewer particles have sufficient track length to achieve a given precision. The 50 nm NP diffuse so little that most trajectories are long enough to be evaluated with a CV of 12% (plateau in a red curve (Fig. 3g). Here, we can analyze 20 NPs in parallel with a CV of 10% for all diameters down to 20 nm. Note that even for the ensemble of the 9 nm NPs, the SEF approach is able to measure a significant number of NPs with long tracks, yielding low CV values (CV = 26% @ $N_p$ = 20, CV = 15% @ $N_p$ = 10, CV = 13% @ $N_p$ = 5). The fast drop-off of the 9 nm ensemble visible in Fig. 3g partially results from the NPs diffusing into the outer regions of the microchannel. In these regions, the NPs cannot be detected, as due to the Gaussian mode shape, the intensity reduces, leading to insufficient scattering signal. Note that the distance between the 9 nm NPs must be larger, as their diffusion covers a much larger volume compared to bigger NPs. Note that with increasing demand for precision, fewer trajectories have the necessary length $N_{f,min}$, leading to a smaller number of NPs to be analyzed.

Independent MSD analysis was performed here in the relevant spatial directions. Figure 4 shows the longitudinal and transversal MSD-curve for all measured gold NP ensembles. While longitudinal

and transversal MSD values are similar on short timespans (Fig. 4b), the transversal MSD deviates from the linear behavior for longer lag times, an effect that is associated with the finite extension of the microchannel restricting the free diffusion (Fig. 4a)[25]. As expected, an earlier onset of the deviation is observed for the smaller NPs due to their faster diffusion. To show this effect more clearly, corresponding simulations are presented in the Supplementary Information (Sec. SI 18 and Fig. S16). Note that each lag time shows the average MSD value of all evaluated NP and does represent a single point in time. The linear fitting of the MSD data points is performed only for the first few lag frames (typical three frames), which have statistically the highest significance and in which no confinement occurs. This allows the diffusion coefficient to be determined independently along the longitudinal and transverse direction[15] (details in SI 2, 4).

The results for all NP ensembles are summarized in Table S1 (more details in SI 5). Note that the hydrodynamic diameter $d_h$ (FaNTA and DLS) is always larger than its physical counterpart $d_p$ (measured by a transmission electron microscope, TEM), because the NPs contain lipoic-dPEG$_{12}$-COOH ligands that are covalently bonded to the particle surface. Note that this acid group provides a highly negatively charged surface that prevents NP aggregation. Overall, FaNTA and DLS measurements are in good agreement, emphasizing the relevance of our approach. A detailed comparison of FaNTA and DLS with respect to the characterization of nanoscale systems can be found in the Supplementary Information (Sec. SI 9, Fig. S-8).

An important question is whether the concentration within the ARE is identical to that of the solution. Using the given channel diameter and fiber length that is microscopically observed ($d_c$ = 17 μm, $l$ = 4096 px = 1420 μm, $V$ = 0.32 nL), we can determine the particle concentration from the measurement. The measured concentration $c_{FaNTA}$ matches the set value $c_{true}$, except for the 9 nm gold NP which are not visible close to the channel wall as described above (see Table 1).

Although the SNR of the image increases with illumination intensity, the total power coupled into the SEF should not exceed a certain limit to avoid a significant effect of photon pressure and

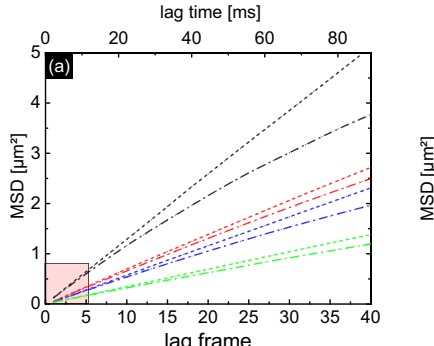
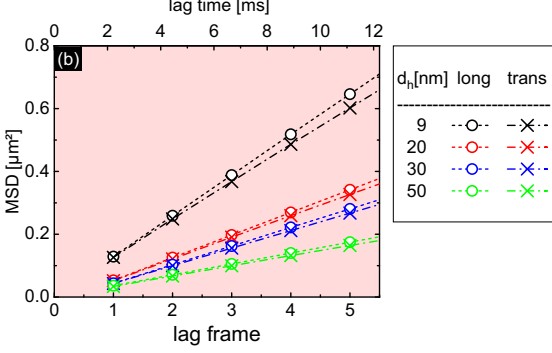

**Fig. 4 | MSD analysis of different ensembles of tracked gold nanospheres.** Each ensemble has a different average hydrodynamic radius (black: 9 nm, red: 20 nm, blue: 30 nm, green: 50 nm). The dashed (dotted dashed) lines refer to the longitudinal (transverse) directions. The left plot **a** shows the analysis of a long time span, while **b** is a close-up view of the first lag times (red background in (**a**)) in which the fitting has been performed.

**Table 1 | Comparison of the main parameters that have been achieved using the SEF in comparison to those measured using DLS**

| Material | unit | 9 nm gold | 20 nm | 30 nm | 50 nm | 50 nm polystyrene |
|---|---|---|---|---|---|---|
| $d_{TEM}$ | nm | 8.8 ± 0.4 | 20.4 ± 0.5 | 28 ± 0.9 | 50.3 ± 2.3 | 51 ± 3 |
| CV | % | 4.4 | 2.5 | 3.1 | 4.5 | 16.5 |
| $d_{DLS, m}$ [manufacturer] | nm | 19 | 28 | 36 | 60 | - |
| $d_{DLS, a}$ [author] | nm | 17.9 | 28.1 | 35.0 | 57.4 | 64.9 |
| $d_{FaNTA-trans}$ | nm | 17.4 ± 0.3 | 27.5 ± 0.6 | 35.7 ± 0.5 | 62.0 ± 1.2 | 64.7 ± 1.1 |
| $d_{FaNTA-long}$ | nm | 16.1 ± 0.1 | 26.5 ± 0.6 | 33.9 ± 0.7 | 59.2 ± 1.6 | 62.8 ± 0.9 |
| $t_{exp-min}$ | μs | 1000 | 500 | 300 | 150 | 400 |
| det. particles (CV = 30%) | | 21 | 38 | 34 | 40 | 13 |
| det. particles (CV = 10%) | | 6 | 20 | 20 | 29 | - |
| $c_{true}$ | NP/nl | 312 | 102 | 120 | 111 | 121 |
| $c_{FaNTA}$ | NP/nl | 65 | 119 | 106 | 125 | 130 |

sample heating. Generally, photon pressure leads to a longitudinal drift of an NP whose average velocity is proportional to the power in the fiber. To clarify this point, the input power coupled to the fiber was varied to determine whether there was an influence on particle drift or on the diameter analysis. It was found that at the laser power required for FaNTA, there was neither significant photon pressure nor relevant heating for all experiments considered in this work. Further details can be found in the Supplementary Information (Sec. SI 6 and SI 7).

## Discussion

The smallest NP diameter measured in this work is 9 nm, which for gold corresponds to a scattering cross-section of 0.026 nm². This is the smallest free-diffusing gold NP detected and individually characterized with record-high precision by NTA that solely uses elastic light scattering. Measuring even smaller NPs is not possible as there are no stable and standardized ultra-uniform NPs available in this size range (SI 8). Note that since interferometric methods (e.g., COBRI[26] and iSCAT[27]) can detect even smaller NPs, we believe that the sensitivity of FaNTA can be improved further by applying an interferometric detection technique in the future.

Compared to our first FaNTA realization relying on nanobore optical fibers[18], the SEF approach allows for reducing the smallest measurable diameter from 19 nm (latex) to 9 nm (gold). Although both types of NPs have a comparable scattering cross-section, the characterization of the 9 nm NPs is substantially more challenging due to twice the higher diffusion coefficient, i.e., a significantly faster movement of the NPs. Additionally, the achieved precision $\sigma_d$ and the number of parallel tracked particles $N_p$ is enhanced 4 to 5 times in this

work. Comparing the 20 nm NP measured here with the 19 nm NP of our previous work[18] we can evaluate 5 times more NP at the same CV, or achieve a four times higher precision for the same number of NPs. In addition, the comparably large microchannel diameter in the SEF prevents the sticking of NPs to the wall, which can be an issue in fibers with nanochannels[18]. These facts clearly emphasize the practical relevance of the SEF approach to reliably detect NPs of extremely small diameter. Note that in case even higher accuracy is required, the sample can be dilute more, allowing longer trajectories and thus accuracies below 1%[19]. To determine the reliability of the measurement system and data analysis on the single NP level, a series of measurements was performed in which the same NP was successively characterized ten times over a defined duration. As the same NP is analyzed, the influence of a size distribution, which would be present for an ensemble, is eliminated. In the present case, a CV of 0.01 is found, which shows the reliability of the combination of measurement system and data analysis. More details on the reliability of the approach (e.g., power stability at the location of the NP) can be found in the Supplementary Information (Sec. S-10 and S-11, Table S-2).

The fundamental diameter limit reached here is 9 nm for gold NP and results from the intrinsic background scattering of water as shown by the following estimation: The probability of a single photon to be scattered on a single 9 nm NP and over the longitudinal distance corresponding to a single pixel (345 nm) is $1.1 \cdot 10^{-10}$ ($\sigma_{scat}/A_{ch} = 0.026\,\text{nm}^2/227\,\mu\text{m}^2$) and $5.9 \cdot 10^{-10}$ ($z \cdot b_{water} = 0.345\,\mu m \cdot 1.7 \cdot 10^{-9}\,\frac{1}{\mu m}$)[28], respectively. Thus, the signal and background have similar intensity. We verified this by using ethanol instead of water, leading to a higher background signal

proportional to its higher scattering coefficient $b_{ethanol}$ (see SI 3). This estimation suggests that a stronger NP signal can be achieved by having a smaller channel diameter. It is important to note that the modal losses in antiresonant fibers scale with $d_c^4$, reducing the illumination intensity significantly when using smaller cores[24]. Further improvements in fiber design will thus make it possible to detect species with even smaller scattering cross-sections in the future. The SEF represents an optimized FaNTA design for long-term stable operation, allowing for the acquisition of long trajectories[23]. A particularly important feature is a single junction between ARE and supporting capillary, allowing for diffraction-limited imaging at comparable low optical losses (see Supplementary Information Sec. SI 13 for details). Future studies will aim to homogenize the mode patterns to reduce mode decay across the channel. Further device integration can be achieved through interfacing SEF with microfluidic circuitry (e.g., as shown here[29]). Here, the dimensions of SEF are particularly suited, as the comparable large core diameter allows for fast liquid exchange[29].

In the present work, experimental investigations and statistical data analysis have been used to investigate the potential of FaNTA with respect to the characterization of very small NPs. By using a novel fiber design, operating through a single ARE, and introducing an optimized chip design, background signals, and aberrations could be efficiently suppressed, allowing deep sub-wavelength species to be characterized at an unprecedented level of precision in the context of NTA. The key result is the detection and characterization of very small NPs with a diameter as small as 9 nm at the CV of only 13%, which corresponds to the smallest NP diameter determined so far with NTA based on pure elastic light scattering at a record low CV, reaching the limit of this type of approach. The study clearly demonstrated that the detectable scattering cross-section within FaNTA is limited only by the background scattering of the ultrapure water, thus reaching the fundamental limit of NTA-technology in general. Influences such as thermal effects or photon pressure could be excluded by detailed experimental investigations.

The individual characterization of NPs with diameters <10 nm at very small CV values represents the current record for elastic light scattering-based NTA. This suggests the relevance of FaNTA in basic science studies (e.g., understanding in situ growth of NPs[1]) or in application-driven fields such as environmental science (e.g., pollution control), semiconductor industry (e.g., monitoring of ultrapure water) or medicine (e.g., development of pharmaceuticals). In addition to the characterization of ensembles of nano-objects, FaNTA uniquely allows analyzing single processes on the nanoscale level, such as the adsorption of molecules at surfaces or changes of conformations, both of which are enabled by the very long observation time through the extraordinarily long observation times.

## Methods

### Dynamic light reference measurements
The dynamic light scattering (DLS) method was used as a reference method to determine the hydrodynamic diameter of the nanoparticle ensembles investigated. For this purpose, DLS instruments were used, which were available in-house (Zetasizer Nano ZS and Ultra; Malvern Panalytical). The sample volume used in each case was 100 μL with a particle concentration of ~100 μg/mL. To increase the measurement accuracy, ten sets of measurements were performed, each consisting of 15 individual measurements. The intensity distribution was used to finally determine the hydrodynamic diameter.

### Data processing
The data processing relies on evaluating the raw data by the method of mean squared displacement (MSD, c.f. SI 2) with TrackPy[30] in combination with a self-written code for parameter estimation[22]. Outliers, coming from contaminations or particle agglomerations, are removed by a $z$-score filtering[31], which operates through the assumption that the determined diffusion coefficients ought to be normally distributed (more details on the $z$-score filtering can be found in the Supplementary Information (Sec. SI 20))[22].

### Reporting summary
Further information on research design is available in the Nature Portfolio Reporting Summary linked to this article.

## Data availability
Data is available upon request from the corresponding author.

## Code availability
Code is available upon request from the corresponding author.

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

## Acknowledgements
This work was supported in part by the German Research Foundation (SCHM2655/15-1 and SCHM2655/21-1). We acknowledge support from the German Research Foundation Projekt-Nr. 512648189 and the Open Access Publication Fund of the Thueringer Universitaets- und Landesbibliothek Jena.

## Author contributions
M.A.S., T.W., and R.F. conceived the study. T.W. and R.F. planned the experiments. J.K. fabricated the ARE fiber. T.W. constructed the measurement setup and carried out the experiments. R.F. created the computer code and analyzed the data. M.N. has done simulations regarding the optical properties of the ARE fiber. The manuscript was primarily written by M.A.S. and R.F. All authors discussed the results and commented on the manuscript.

## Funding

## Competing interests
The authors declare no competing interests.
