## [Peer Review File · Nature Communications]

Characterization of diffusing sub-10 nm nano-objects using single anti-resonant element optical fibersReviewer #2 (Remarks to the Author):

The work reported by Wieduwilt et al. introduces a fiber-assisted nanoparticle tracking method for the characterization of the diameter of nanoparticles within an optical fiber. The key point of this method is to use the diffusion coefficient as a criterion to determine the size of the particles. On the aspect of this point, this work is not new enough because too many works have been performed to characterize the size distribution of the particle with diffusion coefficient. Through confining the particles with an optical fiber, the diffusion areas can be precisely modulated, enabling the detection of small size nanoparticle in the scattering mode. It is worth to point out that the diffusion coefficient of the particles is very sensitive to the property of the solution and surface functionalization. Solution with different viscosity will strongly affect the diffusion behavior. Meanwhile, the chemical properties of the particles will also modulate the interaction with the surface wall of the fiber. In the current form, the results only give a preliminary description on the possibility of this method for particle tracking. More comprehensive experiments are required to further characterize the reliability of this method for size determination. For example, solution with different ionic strength, viscosity and so on. The mixture solution with different size particles. Optimization of fiber size is also required. Particles with different surface functionality also should be discussed. In the presence of confinement effect, the diffusion behavior will be affected. Theoretical simulation results are required. The authors also highlight that this is the smallest free-diffusing gold NP even detected by elastic light scattering. Actually, this description is not acceptable. Many methods have been demonstrated to detect particles with much smaller size. I recommend the authors to further improve the quality of this work. There is a big space to polish the results. In the current form, I think it is much suitable for a more specialized journal with lower quality requirement.

Reviewer #3 (Remarks to the Author):

In the manuscript titled "Characterization of diffusing sub-10 nm nano-objects using single anti-resonant element optical fibers" Wieduwilt and coauthors tested the FaNTA approach for the characterization of very small gold particles, down to the smallest size of 9 nm, with record values for coefficient of variation, approaching the limit of detection of the approach, being limited solely by the scattering of light by the ultrapure water. To achieve this result, they specifically design a fiber that features a single anti-resonant element at one of the distal ends of the hollow channel into which the particles flow. Although the paper has a strong experimental soundness and many aspects that can be defined as "second-order aspects" are well characterized, with only a few experimental "weak points" that can be highlighted, I take the chance to suggest some points that can be improved for better clarity. For this reason, I think the paper can be further considered for publication if the authors can address the concerns listed in the following, in particular concerning differences with their previous works (point 1 below). Apart from this aspect, that I feel should be central in the final decision, firstly, I would like to list the more relevant experimental points, while later in the report I would like to highlight some minor points, mostly related to corrections that can be implemented in the manuscript, figures and other aspects not directly labeled as experimental.

Major points:

1. How far does the proposed paper deviate from the one published by the same authors in August this year? (<https://doi.org/10.1002/sml.202202024>) It seems that the device, the methodology, and the analysis are quite similar between the two papers. Also, the design of the ARE element is quite similar, and in the abovementioned paper more details about the fabrication of the probe and about the data analysis are provided. Although in that case the size of the monitored particles was an order of magnitude bigger (with the smallest detected particles in this work being 9 nm for gold and the smallest particle detected in the bimodal mixture of the August paper being around 60 nm in the supplementary material, while in the main text 100 nm polystyrene beads are analyzed). The authors should provide sufficient details on the methodological and experimental improvements between the two papers, clarifying also in the main text which are the improvements that justify the publication of the reviewed paper in Nature Communications.
2. On page 10 the authors say "Note that with increasing demand on precision, fewer trajectories

have the necessary length $N_{f,min}$, leading to smaller number of NPs to be analyzed". It is not clear if this point is a limitation or a point of strength of the proposed method. This opens to the other question on the statistical analysis, reported also in figure 3.b-e:

2a. Is the number of samples enough to be statistically fitted with the curves used to extract d_p and σ_d ? For Figure 3.d only 11 particles are detected with $N_{f,min}$ of 1000. While the central value of the distribution, which determines d_p , is almost the same for all the figures, the value of σ_d seems dependent on the number of particles detected. The fewer the detected particles, the narrower the histogram is, the smaller the value of σ_d . And if the histograms in figures 3.d and 3.e follow evidently the fitting curve, the same cannot be said for figure 3.b.

2b. Strictly related to point 6.a, how this affects the results in case of a blind experiment in which the mean size of the particle is not already known, as in the case of the experiments reported in the manuscript?

2c. Figures 3.d and 3.e, and partially also figure 3.c show a sort of dip for the bin in between 26 and 27 nm. Could this be related to a bimodal distribution, very close in size, of the analyzed particles?

3. On page 11, the authors state "an earlier saturation is observed for the smaller NPs due to their faster diffusion", but this observation cannot be directly extracted from figure 4 (in particular 4.a), where the curve for 30 nm saturates (or better, suggests the saturation for higher values of lag frames) before 20 nm and the curves for 20 nm and 50 nm show a similar trend. Probably, the x-axis limits can be extended for appreciating more the trend of the curves.

4. On page 15, the authors wrote "Thus, signal and background have similar intensity. We verified this by using ethanol instead of water, leading to a higher background signal proportional to its higher scattering coefficient $b_{ethanol}$ ". It is not clear what has been verified. Did the authors verify only that ethanol produces a higher background signal? Or did they perform a measurement of NPs dispersed in ethanol instead of water? Which is the detection limit in this latter case?

Minor points:

1. In the introduction, the authors state that "The study is based on an optimized experimental configuration and a sophisticated fiber design that consists of one single anti-resonant channel, both of are tailored properties to achieve optimized FaNTA performance" but no details on the optimization, nor a comparison between different types of AREs are provided. How are the characteristics of the ARE chosen? It seems that they are obtained to achieve low modal losses at 532 nm, following Ref. 24, but it could be useful to add those experimental results and simulations to the supplementary material.

2. Is the optimization routine of the butt coupling between the launching system and the fiber channel manual or is it somehow automatized?

3. Are the illumination laser powers predicted analytically for each NP size or are they estimated empirically? If the first is true, which is the analytical relation between P_{min} and d ?

4. Minor: the authors say "Note, that the time a particle can be tracked, can be tuned by the NP concentration". Could the authors elaborate on this point?

5. On page 9 the authors state "Applying the above analysis to all measured NP ensembles confirms, that the CV improves with increasing $N_{f,min}$ ", while Figure 3.f is somehow misleading, being shown in such a way that lets the reader understand that CV is the independent variable, while it depends from N_f .

6. In the introduction, the acronym "NPs" is used in different contexts (in line 3 for Nanoparticles, in line 4 for water contaminants, in line 11, generically to identify nanometric objects). If it is not intended, please use different acronyms for different concepts.

7. References to figure 1.b, 1.c, and 1.d cannot be found in the main text. Please cite them in the main text or move it to the supplementary material (I would suggest citing them in the main text).

8. In figure 1, no caption is written for figure 1.d.

9. Although cited in the text, no labels for the characteristics of the ARE are provided in the SEM micrograph in figure 2. Please add labels and size for w and d_c in figure 2

10. The final sentence in the first paragraph of the methods section should read "which is substantially higher than THE ONE used in the NTA experiments."

11. For figure 3, panel a is cited in the text before panel b.

12. Many references to values that should be in table 1 and in the supplementary information tables are not in there. Examples: d_h , d_p .

13. The acronym TEM for transmission electron microscope is not defined.

14. On page 14, "nanoschannels" should read "nanochannels".

15. In supplementary SI 3, specifically figure S2, I would suggest adding something that can help the reader to have a direct comparison between the background level of water and ethanol. A colorbar, or a small indication of the mean value of the pixels inside the channel could work.

Response letter for our manuscript “*Characterization of diffusing sub-10 nm nano-objects using single anti-resonant element optical fibers*”

Dear Reviewers,

Please find attached the revised version of our manuscript entitled “*Characterization of diffusing sub-10 nm nano-objects using single anti-resonant element optical fibers*” by Torsten Wieduwilt, Ronny Förster, Mona Nissen, Jens Kobelke, and Markus A. Schmidt. We are grateful to the Reviewers for the feedback and for raising important points, which have helped us to improve the manuscript.

In the following, the Reviewer comments are given in black. Please find our responses in blue and revised manuscript text in red (changes are highlighted in red bold). Both in the manuscript and in the supplementary information, the additions are marked in red.

Response to comments of Reviewer #2

The work reported by Wieduwilt et al. introduces a fiber-assisted nanoparticle tracking method for the characterization of the diameter of nanoparticles within an optical fiber. The key point of this method is to use the diffusion coefficient as a criterion to determine the size of the particles.

We would like to thank the Reviewer for the detailed reading of our manuscript and for the comments that helped us to substantially improve the manuscript. Since from our standpoint we were able to address all the issues raised, we sincerely hope that we were capable of improving the manuscript to the Reviewer's satisfaction. The details of our responses can be found below.

R2.1: On the aspect of this point, this work is not new enough because too many works have been performed to characterize the size distribution of the particle with diffusion coefficient.

We would like to thank the Reviewer for raising this important point. It is correct to say

that for the determination of the properties of nanoparticles at the nanoscale, studying the diffusive properties is a widely used approach. In our opinion, this indicates that this is a very attractive field of research, and thus we see our research as a significant contribution in this direction.

To show the difference between published work and the present work and the FaNTA approach as a whole, the main advantages of the FaNTA approach are discussed in the following:

One of the main benefits of the fiber-based approach in comparison to commonly used NTA implementation is the labeling of the individual nano-object by a diameter tag that has a precision that is of the order of the precision provided by the other NTA-technology for the entire ensemble. This is a fundamental difference: NTA gives a precision for the ensemble of nano-objects (mainly resulting from the low number of frames), while FaNTA gives a precision per nano-object. The same is true when comparing FaNTA with dynamic light scatter (DLS) approaches that evaluate the properties of an ensemble of nanoparticles (NPs) rather than individual nanoparticles. Therefore, FaNTA allows for studying individual objects and their dynamics, which cannot be done straightforwardly within the NTA approach or with DLS.

Specifically, the fiber-based FaNTA approach offers the following advantages over commonly used NTA approaches:

- confinement of the NPs in the light-guiding section of the fiber which
 - prevents NPs from leaving the light line illumination beam or the focal plane of the microscope, resulting in exceptionally long observation times, even for very small particles. This represents the key to the high accuracy of the determined diameter of the 9 nm NPs shown in this study.
 - provides a pseudo light line illumination, which yields a very high intensity at the position of the NPs,
 - extends and homogenizes the focal volume for illumination through the translational invariant intensity pattern (i.e., guided mode) along the fiber,
 - reduces the camera readout time, as only a fraction of the entire image needs to be recorded, therefore allowing for high frame rates,
 - leaves sufficient space for measuring the diameter of more than 100 individual NPs simultaneously with a precision that meets medical requirements
- very low signal background from the fiber structure because more than 99.9% of the power of the guided mode is inside the water and not in the glass.
- compatibility with
 - microfluidic thus enabling fast and simple liquid exchange
 - fiber circuitry, allowing for interfacing the system with photonics
- reusability of the individual fiber system
- very small demand on sample volumes (in-fiber volume: $\approx 1nL$)

The specific objective of this manuscript is to demonstrate the potential of the FaNTA approach regarding the smallest possible measurable diameter of a single, individual nanoparticle (not ensemble). Specifically, we demonstrate that FaNTA allows for the characterization of very small NPs (<20nm) at unprecedented levels of precision (record low value of CV = 13%). The smallest diameter being measured is 9nm only,

corresponding to the smallest NP diameter (or scattering cross-section) determined so far with NTA based on elastic light scattering. By using a fiber optimized for FaNTA that includes one tailored anti-resonant element, and a novel chip design, background signals and aberrations could be very efficiently suppressed. Through presenting the limits of FaNTA in terms of photon pressure and heating, our study shows that the detectable scattering cross-section is limited only by the background scattering of ultrapure water, thus reaching the fundamental limit of NTA in general. The achieved results are the current record for elastic light scattering based NTA in characterizing nanoscale species, making the FaNTA-platform highly attractive for basic science studies (e.g., understanding in-situ growth processes of nanoscale species) or in topical applications such as environmental science (e.g., pollution control), semiconductor industry (e.g., monitoring of ultrapure water) or medicine (e.g., development of pharmaceuticals). Note that FaNTA also allows analyzing single nanoscale processes on the individual NP level due to the very long observation time. Due to the unique features and potential of the presented study and the FaNTA approach in general, we strongly believe that this work will create substantial impact in the Photonics, Bioanalytics and Nanoscience communities.

At this point in the response letter, we would like to explicitly point out the key differences between FaNTA and DLS to make it clear that FaNTA represents a novel analytical approach for the characterization of nanosystems.

A. Level of NP concentration: To achieve sufficient scattering intensities, DLS demands very high concentrations of the NPs in the solution under investigation. Note that very small NPs with small scattering cross-section may require NP densities at which undesirable interparticle interactions occur. Note that achieving very high concentrations (about $10^{11} \dots 10^{14} \text{ NP/mL}$), which might actually be higher than that of the stock solution used, may require additional experimental steps such as centrifugation. Such steps, however, should be avoided for sensitive biological systems. FaNTA, in contrast, operates at substantially lower concentrations, as individual NPs are analyzed, thus preventing interparticle interaction and allowing to investigate the individual NP unambiguously.

B. Knowledge in NP concentration: Another important distinguishing feature is that within DLS, the temporal behavior of the scattered intensity is evaluated. A consequence of this approach is that the determined hydrodynamic diameter of the relevant NPs depends on the concentration of the solution investigated. Thus, a concentration series must be performed in order to determine the correct diameter. To demonstrate this behavior in the context of the study discussed here, DLS-related experiments using on our in-house DLS-instrument (Zetasizer Nano ZS and Ultra; Malvern Pananalytical) were performed by successively diluting a highly concentrated solution of gold NP (physical diameter 50nm) and characterizing it in the Zetasizer (Fig. RL1(a), Fig. S-8 (a) in the supplementary information). In accordance with the discussion above, a very strong concentration dependence was found. For comparison, FaNTA experiments were performed with a specific solution (diluted with 0.1% TWEEN20, concentration 10^{-8} NP/mL), which revealed an average diameter of 64.3nm (Fig. RL1(b), Fig. S-8 (b) in the supplementary information). This clearly shows that diameter determination in DLS always requires a concentration series, which is

not necessary when using FaNTA.

Fig. RL1 (Fig. S-8 in the supplementary information): Comparison of DLS and FaNTA regarding diameter determination of NPs. (a) Concentration series measured by using our in-house DLS machine showing the concentration dependence of the determined hydrodynamic diameter being intrinsic to DLS (physical diameter 50nm) (top axis: number concentration. Bottom axis: mass concentration (density)). The dots refer to the mean diameter of the respective DLS measurement, where each measurement was realized by multiple dilution of the initial stock solution (gray dot). The light yellow dot shows a reference measurement at the concentration yielding the largest hydrodynamic diameter after four days, confirming the reproducibility of the measurement principle. (b) Results of the corresponding FaNTA measurements. Shown is the diameter distribution as a function of trajectory length. The color bar refers to the respective trajectory. The resulting mean hydrodynamic diameter is 64.3nm (horizontal red dashed line).

These results clearly show that certain assumptions must be generally considered in case a DLS-based analysis is used, whereby the results cannot be interpreted unambiguously. In contrast, FaNTA requires the intensity only for NP localization and not for the actual determination of the hydrodynamic diameter, since FaNTA solely analyzes the trajectory of the individual NPs. Thus, no concentration dependence of the diameter to be determined is present for FaNTA, which clearly reveals that FaNTA is fundamentally more independent of any initial assumptions.

C. Refractive index of solution: Another consequence of the intensity being key for the data evaluation in DLS is that accurate knowledge of the refractive index (RI) of both the NP and the solution is required to convert the intensity distribution to number and volume distribution. The RI, however, remains unknown for many NP systems as well as fluids and is particularly critical to define for composite nano-objects such as core-shell NPs. As mentioned, since FaNTA has no direct intensity dependence, no dependence of the diameter determination on refractive index of NP and solution has to be considered, emphasizing the independence of FaNTA from initial assumptions.

D. Ensemble vs. individual NP characterization: Another advantage of FaNTA compared to DLS lies in the ability to label the individual NP with a highly precise

diameter tag. This constitutes a significant distinction between the two approaches: while DLS provides precision at the ensemble level, FaNTA offers high precision per individual NP. As a result, FaNTA enables studying of dynamics at the single NP level, which cannot be done by DLS. Note that due to the exceptionally long trajectories and the very high statistical significance achieved, FaNTA uniquely allows discriminating bimodal solutions where the two ensembles have close mean diameters. For example, one of our recent works has shown that FaNTA is able to discriminate a mixture consisting of two polymer NPs ensembles with very close mean diameter (125nm and 100nm), with DLS failing in this situation [Nis2022].

E. Actions taken

To account for the Reviewer's point, we have improved the argumentation in the manuscript as follows:

Introduction: ... One approach applied in this context is dynamic light scattering (DLS), evaluating the diffusive properties of ensembles of NPs. Here, even though NPs such as proteins or molecules can be characterized [REF], polydisperse samples are difficult to handle, very high and partially inappropriate specimen concentrations are required, and larger NPs are intrinsically preferred [REFs]. **Moreover, DLS does not allow analyzing the dynamics of NPs at the level of individual objects, since the properties of an ensemble of NPs and not those of individual particles are determined.** ...

Introduction: ... The key feature of FaNTA is the confinement of NPs in fiber-integrated microchannels. It leads to high-intensity pseudo light line illumination through the optical mode **and fast readout times. Particularly relevant is confinement of the NPs to the light-guiding sections, preventing NPs from leaving the illuminated volume or the focal plane of the microscope. This results in exceptionally long observation times (i.e., trajectories with very large number of frames) of rapidly diffusing objects, leading to high statistical significance in diameter determination of individual NPs, even for very small particles [REFs]. Note that in other NTA implementations, precision is defined with respect to the entire NP ensemble (mainly resulting from the low number of frames), while FaNTA gives a precision per individual NP.** ...

Introduction: ... In this work, we ~~uncover~~ **unlock** the potential of FaNTA for the characterization of very small NPs (<20nm) by means of experimental investigations and detailed statistical analysis (Fig. 1(a)). The study is based on an optimized experimental configuration and a sophisticated fiber design that consists of one single anti-resonant channel, both ~~of~~ **are having** tailored properties to achieve optimized FaNTA performance. **Based on the results of our previous work that concentrated on the microscopic imaging through the ARE and the capabilities to separate bimodal mixtures [Nis2022], we focus here on uncovering the potential of the ARE-concept with respect to the characterization of NPs with extremely small diameters, where we were able to uncover the fundamental limits and influences. In addition to the aspects addressed in [Nis2022] we discuss in this work issues such as the impact of the microchannel confinement on diffusion and data analysis, the realization of a novel chip design, or the influence of**

photon pressure, and present an experimental reliability study of the ARE-concept. All the issues addressed in this study are related to the characterization of very small NPs. The key result is the characterization of diffusing NPs as small as 9 nm with record low CV values (CV = 13%, **an example of tracking such small NPs with FaNTA is shown in Figs. 1 (b) to (d)**). The detectable scattering cross-section is here only limited by the background scattering of the ultrapure water, thus reaching the fundamental limit ~~in the context~~ of this approach. ...

Abstract: ... Accurate characterization of diffusing nanoscale species is increasingly important for revealing processes at the nanoscale, with fiber-assisted nanoparticle tracking analysis (FaNTA) representing a new and promising approach in this field. In this work, we uncover the potential of FaNTA for the characterization of very small NPs (<20nm) through experimental studies, statistical analysis and the employment of a sophisticated fiber and chip design. The central result is **the characterization of diffusing NPs as small as 9 nm with record-high precision, corresponding to the smallest diameter yet determined for an individual nanoparticle with NTA using elastic light scattering alone**. Here, the detectable scattering cross-section is limited only by the background scattering of the ultrapure water, thus reaching the fundamental limit **of Nanoparticle-Tracking-Analysis in general**. The obtained results outperform other NTA realizations and allow access to previously difficult to address application fields such as understanding nanoparticle growth or control of pharmaceuticals. ...

Main text: ... Overall FaNTA and DLS measurements are in good agreement, emphasizing the relevance of our approach. **A detailed comparison of FaNTA and DLS with respect to the characterization of nanoscale systems can be found in the Supplementary Information (Sec. S-9, Fig. S-8).** ...

Discussion: ... The smallest NP diameter measured in this work is 9 nm, which for gold corresponds to a scattering cross-section of 0.026 nm². **This is, to the best of our knowledge, the smallest free-diffusing gold NP detected and individually characterized with record-high precision by NTA that solely uses elastic light scattering.** ...

Conclusions / outlook: ... By using an optimized fiber design, operating through a single ARE, and introducing a ~~optimized~~ **novel** chip design, background signals and aberrations could be efficiently suppressed, allowing deep sub-wavelength species to be characterized at **an** unprecedented level of precision in the context of NTA. The key result represents the characterization of very small NPs with a diameter as small as 9nm at CV of only 13%, corresponds to the smallest diameter determined so far using elastic light scattering based NTA at a record low CV. The study clearly demonstrated that the detectable scattering cross-section within FaNTA is limited only by the background scattering of the ultrapure water, thus reaching the fundamental limit of **NTA-technology in general.** ...

Furthermore, we have also supplemented the Supplementary Information with the parts describing FaNTA and DLS (see Sec. SI 9).

Overall, we hope that with this discussion we have been able to present in detail the main features and advantages of FaNTA and convince the Reviewer of the potential of this technology in terms of characterization of very small diffusing NPs.

R2.2: Through confining the particles with an optical fiber, the diffusion areas can be precisely modulated, enabling the detection of small size nanoparticle in the scattering mode.

We thank the Reviewer for the comment, to which we fully agree. By the employment of the light-guiding microchannel, the NPs are confined to the field-of-view, allowing for extraordinarily long observation times. This ultimately results in trajectories that contain a very large number of frames, leading to exceptionally high accuracy in the statistical data analysis, i.e., the determined NP diameter. Overall, as mentioned above, FaNTA gives a precision per nano-object, while other NTA and DLS implementations define precision with respect to the ensemble of nano-objects (mainly resulting from the low number of frames). This unequivocally demonstrates the uniqueness of the FaNTA approach and of the ARE-fiber used in this work.

To account for the Reviewer's comment, we have changed the text according to the changes mentioned in our response to the first comment of the Reviewer (R2.1).

R2.3: It is worth to point out that the diffusion coefficient of the particles is very sensitive to the property of the solution and surface functionalization. Solution with different viscosity will strongly affect the diffusion behavior.

Also, in this case the Reviewer is correct. We first like to comment on the dependence of the viscosity and then on the surface functionalization.

A. Diffusion properties: The diffusive behavior of the NPs is very much dependent on the properties of the solution, especially on the viscosity. One decisive factor here is the temperature dependence of the viscosity $\eta = \eta(T)$, which is related to the hydrodynamic radius to be determined R_h through the Einstein-Stokes equation.

$$D = \frac{k_B T}{6\pi \cdot \eta(T) \cdot R_h}$$

Here, water shows a very pronounced temperature dependence of viscosity, as shown for example in the work of Ye *et al.* [Ye2012] (Fig. RL2).

Fig. RL2: Temperature dependence of the viscosity of water measured by Ye *et al.* [Ye2012].

The influence of temperature in relation to the experiments discussed here represents an important point of the study shown in this manuscript. Here we refer to the

Supplementary Information of our work, in which we discuss the influence of light-induced heating of the NPs. In detail, power-dependent measurements were performed, which clearly showed that under the measurement conditions used, no temperature-dependent influence has to be considered. For further details, we would like to direct the Reviewer's attention to Sec. SI 7 of the Supplementary Information.

To make this point more clear, we have changed the main text of the manuscript as follows: ... Generally, photon pressure leads to a longitudinal drift of a NP whose average velocity is proportional to the power in the fiber. **To clarify this point, the input power coupled to the fiber was varied to determine whether there was an influence on particle drift or on the diameter analysis. It was found that at the laser power required for FaNTA, there was neither significant photon pressure nor relevant heating for all experiments considered in this work. Further details can be found in the Supplementary Information (SI 6 and SI 7). ...**

B. Surface functionalization: We fully agree with the Reviewer on this aspect. A functionalization of the surface of NPs can substantially alter their diffusive properties and thus the determined hydrodynamic diameter. In this context, we have performed NTA-experiments in a different waveguide system with substantially larger NPs to reveal the capability of FaNTA for measuring the impact of changes of the morphology of NPs [Kim2022]. In detail, we have demonstrated the relevance of waveguide-based NTA by measuring the solvency-induced collapse of a NP system, which includes polymer brush-based shells as surface functionalization that react to changes in the liquid environment (Fig. RL3). Via changing the liquid environment (case 1: water, case 2: mixture of ethanol and water), we found that under good solvency conditions (in pure water) the polymer shell was swollen, while the addition of ethanol imposes a reversible collapse of the polymer shell. Kindly note that as shown in this study, DLS reference measurements (with an in-house Zetasizer) lead to false outputs and only the waveguide-NTA-approach provides meaningful results.

Fig. RL3: Demonstration of the capabilities of the waveguide-based NTA approach for characterizing NP ensembles that include a surface functionalization that reacts to changes of the environment. Histogram representation of the diameter distribution of gold NPs with (PNiPAAM) shells in the solvent-swollen state (green), and after co-solvency induced shell collapse (purple) ((a) 30 kDa, (b) 9 kDa). The tracking was conducted in the xz-plane. More details can be found in [Kim2022].

Since in this manuscript we have limited the scope of this study to the optimization of the FaNTA system with respect to reveal the smallest possible NP diameter that can be accurately measured, we have deliberately selected NPs with defined and inert surfaces from companies providing well-established NPs. Since these NPs are pre-characterized, surface-related effects can be excluded and the study can focus exclusively on the aforementioned target.

C. Actions taken

To emphasize this point in the manuscript, we have made the following change in the main text: ... To unlock the limits of the FaNTA realization discussed here, we imaged and analyzed 9, 20, 30 and 50 nm ultra-uniform gold and 50 nm latex particles (**details of the NPs can be found in Tab. S-1 of the Supplementary Information**). **Note that we deliberately selected NPs with defined and inert surfaces from companies providing well-established NPs to avoid surface-related effects which may impact diffusion.** The ideal specimen concentration is around 100 NP/nL, depending on the particles' diffusion coefficient and scattering cross-section. ...

R2.4: Meanwhile, the chemical properties of the particles will also modulate the interaction with the surface wall of the fiber.

Again, we agree with the Reviewers that the chemical properties of the NP can influence its interaction with the surface of the microchannel. Regarding the experiments discussed here, we would like to mention that in all gold-NP based FaNTA experiments we have performed so far, no surface interaction could be observed. Here it is worth mentioning the experiments that use fibers with much smaller channels (e.g., nanobore fiber with channel diameter of only 400nm [Gui2021, Fae2015]), in which also no interaction with the wall could be observed, although the NPs hits the wall much more often than in the ARE fiber used here (channel diameter: 17 μ m). Since fibers with comparable diameter are used in the context of the manuscript discussed here, an influence of a surface interaction can be excluded.

To account for the Reviewer's point, the text of the manuscript has been amended as follows: ... Outliers, coming from contamination or particle agglomerations, are removed by a z-score filtering [REF], which operates through the assumption that the determined diffusion coefficients ought to be normally distributed [REF]. **Kindly note that an interaction of the NP with the wall of the channels could not be observed in any of the previous experiments using gold NPs (even in fibers with much smaller fluidic channels [Gui2021, Fae2015]). Thus, an influence of a surface interaction on the diameter determination can be excluded in the experiments discussed in the following.** ...

R2.5: In the current form, the results only give a preliminary descript on the possibility of this method for particle tracking. More comprehensive experiments are required to further characterize the reliability of this method for size determination.

We thank the Reviewer for the remark regarding the reproducibility of the approach presented here. The reproducibility of the FaNTA approach depends on various issues as outlined in the following:

A. fiber geometry

An important factor for the reliability of FaNTA is the preservation of the structural parameters of the fiber cross-section along the fiber. Specifically, in the experiments discussed here, maintaining the dimensions of the ARE channels is crucial, which must be identical (i) within a sample and (ii) from sample to sample. In this context, one of the most significant advantages of *Fiber Optic Technology* comes into play: The implementation of optical fiber has reached a level that hundreds of meters of fibers with identical cross-sections at any longitudinal position can be drawn in one go. This aspect, which is also valid in the experiments presented in this work, yields an overall very high reproducibility of the FaNTA approach. Note that the extraordinary reproducibility of optical fibers is one of the key features of fiber optics that actually has enabled today's telecommunications. In the samples used, no measurable change in the fiber cross-section has been found, thus no variation of the fiber parameters has to be considered.

B. Stability of light transmission

An example of the time evolution of the power transmitted through the water-filled microchannel (normalized to the average value $P_n = P/P_{ave}$) over a period of 30 min is shown in Fig. RL4 (Fig. S-9 of the supplementary information). Experimentally, the light originating from the microchannel at the fiber output was imaged onto a Si-photodetector by means of a lens and a pinhole. A negligible variation in output power over this period is observed (standard deviation $\sigma_p = 0.0045$), demonstrating the stability of intensity at the location of the NPs over a time period that is substantially longer than that of the experiments discussed here. Note that the power variations are largely caused by the butt coupling between the delivery fiber and the ARE-fiber, which can be principally improved by an optimized design of the sample mount.

Fig RL4 (Fig. S-9 in the supplementary information): Normalized transmitted power at the output of the water-filled ARE as a function of time over a period of 30 minutes. The data is normalized to the mean value of the output power (yellow dashed line). The green area represents the range corresponding to a width of the first standard deviation ($P_n = P_{ave} \pm \sigma_p$), i.e., the range in which 68.2% of the power values fall. The standard deviation is $\sigma_p = 0.0045$.

C. Single nanoparticle reliability of the measurement system and data analysis

An important parameter for revealing the reliability is the coefficient of variation (CV), which is basically defined as the ratio of standard deviation and mean value. In the case of NTA, the CV is generally defined by $CV_M = \sqrt{CV_E^2 + CV_S^2}$ with the measured CV CV_M and the CVs of the NP ensemble and the system CV_E und CV_S . Thus, it can be inferred that the parameter CV_S can serve as an indicator for the reliability of the measurement system and the associated data analysis. In order to eliminate the influence of the size distribution of an NP ensemble, i.e. the influence of CV_E , a series of measurements was carried out in which the same NP was successively characterized 10 times. Thus, $CV_E = 0$ and the measured CV characterizes the reliability of the system ($CV_M = CV_S$). The resulting values for the determined mean diameter with the corresponding CV including the measurement conditions used are summarized in the following table (Tab RL1, Tab. S-2 in the supplementary information).

Tab RL1: Results of reliability study, using 10 individual sequential measurements of the same gold nanopshere (nano Compositex) within a total period of $T = 15min$. The measurement conditions are as follows: frame rate $\nu = 450fps$, image size: $(80 \times 4096) pixel$ ($(0.028 \times 1.42)mm$), exposure time $\tau = 1.9ms$, time of each measurement: $t_m = 20s$ (9000 images), time between measurements $\Delta t_m \approx 1.5min$.

measurement no.	median [nm]	average [nm]	CV
1	54.8	53.1	0.059
2	52.5	52.3	0.036
3	53.3	53.1	0.048
4	52.8	53	0.056
5	51.7	52.8	0.095
6	51.8	52.1	0.063
7	52.6	52.3	0.06
8	53.6	53.7	0.058
9	50	51.5	0.066
10	54.1	54.3	0.026

The data analysis yields a hydrodynamic diameter averaged over all ten measurements of $d_{all} = 52.82nm$ with a standard deviation of $\sigma_{all} = 0.77nm$, resulting in a system-related CV of $CV_S = 0.01$. This low value clearly confirms the excellent single NP reliability, i.e. the reliability of the combination of measurement system and appropriate data analysis and its potential for the characterization of nano-objects in general.

D. summary and actions taken

To account for the Reviewer's comment, the reliability studies addressing the power stability of the transmitted light (Sec. B) and the 10 successive FaNTA measurements (Sec. C) have been included into the Supplementary Information (Sec. S-10 and S-11). The main text in the manuscript has been extended as follows:

... These facts clearly emphasize the practical relevance of the SEF approach to

reliably detect NPs of extremely small diameter. Note that in case even higher accuracy is required, the sample can be **diluted** more, allowing longer trajectories and thus accuracies below 1%. **To determine the reliability of the measurement system and data analysis on the single NP level, a series of measurements was performed in which the same NP was successively characterized 10 times over a defined duration. As the same NP is analyzed, the influence of a size distribution, which would be present for an ensemble, is eliminated. In the present case a CV of 0.01 is found, which shows the reliability of the combination of measurement system and data analysis. More details on the reliability of the approach (e.g., power stability at the location of the NP) can be found in the Supplementary Information (Sec. S-10 and S-11, Tab. S-2). ...**

R2.6: For example, solution with different ionic strength, viscosity and so on. The mixture solution with different size particles.

Again, we thank the Reviewer for this comment. Since the first two points have been addressed in our previous responses, we kindly refer the Reviewer to our responses to the third and fourth comments (R2.3 and R2.4).

In the context of mixed solution containing different NP ensembles, we would like to refer to one of our recent works [Nis2022]. There we showed that FaNTA allows to separate two NP ensembles with very close average diameters in a bimodal solution, which is not possible with DLS. In detail, we experimentally demonstrate that FaNTA allows resolving key properties of bimodal polymer NP mixtures with very close mean diameters (100 nm and 125 nm) at record high separation indices ($S = 0.7$, green bars and orange curve in Fig. RL5). Such a bimodal mixture is inseparable using a commercial DLS device (measured in-house, gray bars in Fig. RL5), and can be hardly characterized through other NTA realizations that typically have much lower S values. For more details, we kindly refer the Reviewer's attention to our published work in Small [Nis2022].

Fig. RL5: Measured hydrodynamic diameter distribution of a bimodal solution using FaNTA, continuing two types of polystyrene NP ensembles with very close mean diameter (100nm and 125nm). Green: Histogram. Orange: fitted Gaussian mixture model. The gray bars in the background show the results of the corresponding DLS measurements. The presented results are taken from a previous publication [Nis2022].

Since this study on the separability of bimodal solutions using FaNTA has already been published, we refrain from including this important aspect into the current manuscript and hope that it is ok for the Reviewer based on the above reasoning. Note that the aforementioned work is already cited several times in the manuscript.

R2.7: Optimization of fiber size is also required.

We thank the Reviewer for the remark regarding the optimization of the fiber parameters, which was also mentioned by Reviewer #3 in her/his fifth comment (R3.5). The fiber design was optimized from various perspectives, which we individually outline in the following:

A. Optimization of spectral properties: Crucial to the optical behavior of this type of fiber is the light guiding mechanism within the ARE, which in the case of the fiber used here is the anti-resonant effect. Since this effect relies on interference, transmission bands that are limited by strong resonances emerge in the spectral distribution of the transmitted power. To spectrally locate the operation wavelength ($\lambda = 532\text{nm}$) into one of the transmission bands (i.e. low-loss region), an appropriate wall thickness of the silica-based ARE was chosen during fiber fabrication ($w = 730\text{nm}$). To emphasize that the wall thickness dependence is essential, we have calculated the distribution of the modal attenuation of the fundamental mode in an annulus having the same diameter as the ARE used in the experiments at the operation wavelength for the air (red) and water (blue) cases (Fig. RL6, Fig. S-10 in the supplementary information):

Fig. RL6 (Fig. S-10 in the supplementary information): Simulation of modal loss of the fundamental core mode of an annulus waveguides (material sequence: liquid/silica/liquid) as a function of wall thickness at the operation wavelength ($\lambda = 532\text{nm}$). The core diameter was chosen to match the diameter of the ARE used in the experiment ($17\mu\text{m}$). The two colors refer to water (blue) and air (red).

The plot clearly shows that a precise choice of wall thickness is required to reach the condition of lowest loss within one transmission band. This point was already mentioned in the submitted version of the manuscript and has been slightly extended with Fig. RL6 now included in the supplementary information: ... The wall thickness ($w = 730 \pm 20 \text{ nm}$) of the ARE element (diameter $d_c = 17 \pm 0.5 \mu\text{m}$) was chosen such to yield low modal losses (0.4 dB/cm in simulation and experiment, see Supplementary

Information Sec. SI 12 (Fig. S-10) for further details) at the operation wavelength ($\lambda = 532\text{nm}$). **More details on defining the spectral operating range can be found in this work [Zei2017]. From a waveguide and imaging perspective, the presented fiber design has been optimized with respect to several aspects, including the effect of the ARE-jacket junction, modal losses, or the size of the microchannel (for more details, see the Supplementary Information (Secs. SI 13 and SI 14, Figs. S-11 and S-12)). ...**

B. Optimization of imaging properties: An essential prerequisite for using FaNTA to track small NPs is aberration-free microscopic imaging of the NPs. The presence of ideal imaging conditions is crucial to reduce localization inaccuracies as much as possible and to avoid inaccuracies of the trajectories.

For this purpose, the single ARE fiber used in this study was developed, which allows imaging through the sintering point of ARE and cladding, thus minimizing the number of interfaces between NPs and microscope objective. This is a significant advantage over conventional hollow-core fibers, where the light scattered by the NPs has to pass through a large number of interfaces, resulting in aberrations.

As shown in our previous work, we found that the imaging through the junction is almost aberration free, as suggested by a cross-correlation analysis that analyzes the recorded image of a single NP in relation to the ideal point spread function (see Fig. RL6 and Figs. S6, S7 and S8 of [Nis2022]). Furthermore, the core diameter was adjusted to the depth of field of the microscopic objective to obtain a sharp image over the entire core diameter.

Fig. RL6 (Fig. S7 from [Nis2022]): Results of the cross-correlation analysis, comparing the recorded image with the appropriate ideal point spread function. (a) Peak value of the normalized cross-correlation function along a full FaNTA video for two exemplary NPs. The pixel value in the legend gives the NP position along the fiber in the first frame. (b) Same plot with magnified y-scale. For more details please see Fig. S7 of the Supplementary Information of [Nis2022].

To account for the Reviewer's comment, the main text of the manuscript has been amended by the following section: ... **The main advantage of this arrangement is the single connection point of the ARE with the supporting capillary (Fig. 1(a)), yielding excellent tracking properties and nearly aberration-free imaging as the number of disturbing interfaces between NP and microscope objective is reduced to a**

minimum. As shown in [Nis2022] by a cross-correlation analysis correlating the recorded image of a single NP with the ideal point spread function, the resulting image is diffraction limited for almost all transverse positions of the NP. ...

C. Optimization of modal attenuation: One crucial aspect that needs to be considered in case anti-resonant waveguides are employed are the modal losses, which need to be sufficiently low to ensure a constant light intensity longitudinally over the entire field-of-view. The main loss distribution in the present type of anti-resonant waveguide is the sintering point between ARE and cladding, which intrinsically has low reflectivity due to the absence of interference between two interfaces. To illustrate this effect, the influence of the geometry of the ARE-jacket junction was simulated via FEM by modeling the modal losses of the core mode at the operation wavelength ($\lambda = 532\text{nm}$, water-filled case). Note that the azimuthal asymmetry of the ARE cross-section results in birefringence, which removes the degeneracy of the fundamental mode of the ideal cylindrical structure, leading to modal splitting of the fundamental mode. As shown in Fig. RL7(a), the modal loss increases as the width of the junction is increased. To find a compromise between losses and size of the contact point, a junction width of $w = 2\mu\text{m}$ (Fig. RL7(d)) was chosen in the experiments reported here. The corresponding simulations showed losses of $\gamma = 0.4\text{dB/cm}$, yielding a reduction of the power transmission over the length of the field-of-view ($L = 1.4\text{mm}$) of $\Delta T = 1.3\%$. This reduction is practically negligible and thus a constant illumination of the NP across the entire measurement area can be assumed.

Fig. RL7 (Fig. S-11 in the supplementary information): Impact of the junction between the ARE and the cladding on the modal attenuation of the fundamental ARE modes at the operation wavelength ($\lambda = 532\text{nm}$, water-filled case). (a) Modal attenuation as a function of the width of the junction of the two fundamental modes. The inset on the top left defines the width parameter w . The intensity distributions of the two modes (h: horizontal, v: vertical) for $w = 2\mu\text{m}$ are shown in the lower right insets (white arrows: direction of electric field at a fixed point of time). The different configurations simulated are shown in images on the right ((b) $w = 10\mu\text{m}$, (c) $w = 6\mu\text{m}$, (d) $w = 2\mu\text{m}$, (e) $w = 0.1\mu\text{m}$, (f) $w = 0\mu\text{m}$). The configuration framed in red (configuration (d)) refers to the structure used in the experiments.

D. Measurements of modal attenuation

Experimentally, the losses of a water-filled ARE fiber were determined by the cut-off method using successive shortening of the sample length and subsequent power transmission measurement (Fig. RL8). Linear fitting (in log-scale) of the measured data results in a modal loss of $\gamma_{exp} \approx 0.4 \text{ dB/cm}$, which agrees well with simulated data of an air-filled ARE fiber ($\gamma_{sim}(w = 2 \mu\text{m}) = 0.4 \text{ dB/cm}$). This agreement shows (i) the high accuracy of the fiber implementation procedure and (ii) that the simulated-based designs can precisely be transferred into real-world fibers.

Fig. RL8 (Fig. S-12 in the supplementary information): Transmitted power through the ARE as a function of sample length in case the fiber is filled with air. The dashed line corresponds to a linear fit to the data point, yielding losses of 0.39 dB/cm at the operation wavelength ($\lambda = 532 \text{ nm}$).

This point has been included into the manuscript as follows: ... The wall thickness ($w = 730 \pm 20 \text{ nm}$) of the ARE element (diameter $d_c = 17 \pm 0.5 \mu\text{m}$) was chosen such to yield low modal losses (0.4 dB/cm in simulation and experiments [REF], see **Supplementary Information Sec. SI 12 (Fig. S-10) for further details**) at the operation wavelength ($\lambda = 532 \text{ nm}$). **More details on defining the spectral operating range can be found in this work [Zei2017]. From a waveguide and imaging perspective, the presented fiber design has been optimized with respect to several aspects, including the effect of the ARE-jacket junction, modal losses, or the size of the microchannel (for more details, see the Supplementary Information Secs. SI 13 and SI 14, Figs. S-11 and S-12))...**

To account for the Reviewer's comment, we have included the discussions related to the simulations of the impact of the junction (Sec. C) and the experimental attenuation data (Sec. D) into the Supplementary Information (see Secs. SI 13 and SI 14, Figs. S-11 and S-12).

In light of the new results, the following text has been modified: ... A particular remarkable feature is the single junction between ARE and supporting capillary, allowing for diffraction-limited imaging **at comparable low optical losses (see Supplementary Information Sec. SI 14 for details)**. Future studies ~~will aim to understand the impact of this junction on modal attenuation and~~ will aim to homogenize the mode patterns to reduce the mode decay across the channel. ...

In summary, the above discussion shows that in this study addressed in this work, a

fiber design optimized for the particularities of FaNTA can be experimentally implemented by fiber fabrication.

R2.9: Particles with different surface functionality also should be discussed.

Here we would like to direct the Reviewer's attention to our response to her/his third comment (R2.3, Sec. "B. surface functionalization") in which we describe FaNTA-based experiments that show a characterization of a surface sensitive system. In detail, we have measured the solvency-induced collapse of a NP system, which includes polymer brush-based shells as surface functionalization that react to changes in the liquid environment. The paper published in ACS Sensors can be found here [Kim2022].

R2.10: In the presence of confinement effect, the diffusion behavior will be affected. Theoretical simulation results are required.

Here we would like to mention that this point has been addressed in the manuscript, we: ... While longitudinal and transversal MSD values are similar on short timespans (Fig. 4 (b)), the transversal MSD saturates for longer lag times, an effect that is associated with the finite extension of the microchannel restricting the free diffusion (Fig. 4(a)). As expected, an earlier saturation is observed for the smaller NPs due to their faster diffusion. Note that each lag time shows the average MSD value of all evaluated NP and does represent a single point in time. The linear fitting of the MSD data points is performed only for the first few lag frames (typical 3 frames), which have statistically the highest significance and in which no confinement occurs. This allows the diffusion coefficient to be determined independently along the longitudinal and transverse direction (details in SI 2 & 4). ...

In accordance with the Reviewer's comment, we would like to present in the following several simulations, in which the influence of confinement on the diffusion of NPs having diameters close to those used in the experiment is shown. We first discuss the resistance factor, which describes the influence of confinement on the diffusion (i.e., viscosity) itself, and then reveal the impact on the data analysis, i.e. on the MSD-lag time dependence.

A. Resistance factor

At this point, we would like to mention that we have already performed experiments with fibers containing much smaller fluidic channels than the ones used in this work, indeed showing a confinement effect. For example, we studied the diffusion of a 50nm gold NP inside a modified graded index fiber that had a channel diameter $>1\mu\text{m}$, showing a clear impact of the confinement [Jia2020].

The physics behind the confinement effect relies on the fact that the viscosity of the liquid can no longer be considered constant when the NP approaches the wall due to boundary effects. As a result, the viscosity increases towards the wall, which can have an influence especially in the case of small channel diameters and large NPs. Typically, this confinement-related effect is summarized in spatially dependent the resistance coefficient $R_q(x, y)$. This parameter is commonly averaged by different models, resulting in the spatially invariant resistance factor R_s , which summarizes the confinement effect in a single quantity and ultimately defines a new viscosity ($\eta = R_s\eta_0$,

η_0 : bulk viscosity). According to the existing literature, the model of H. Brenner and L. J. Gaydos [Bre1977] has been established, which has been extended to a larger parameter range by J. M. Nitsche and G. Balgi [Nit1994]. In the following, the latter model, which describes diffusion inside an infinitely extended cylinder, will be used to unravel the influence of the confinement effect on diffusion along the cylinder (i.e., fiber) axis (longitudinal direction). In all models, the crucial parameter is the ratio between the radii of the NP and the channel $\lambda = a/R$ (a : radius of NP, R : radius of channel). The symbol λ was chosen here in accordance with the work of [Nit1994]. This dimensionless parameter allows calculating the resistance factor independent of the actual geometric dimensions of the system investigated.

The most important parameter in the work of J. M. Nitsche und G. Balgi [Nit1994] is the diffusion correction factor k (Eq. 45 of the mentioned work) which is determined for various combinations of NP and channel radii (Fig. RL9(a)). This parameter is inverse proportional to the resistance factor ($k = 1/R_s$) and describes how the longitudinal free diffusion coefficient is changed by the confinement ($k = \tilde{D}/D_\infty$; measured and free diffusion coefficients: \tilde{D} and D_∞). Clearly visible is a substantial deviation of the diffusion correction factor from unity, particularly in the case of small channel radii and large NPs, i.e., large values of λ . This effect can also be seen in Fig. 2 of the work of J. M. Nitsche und G. Balgi [Nit1994] (Fig. RL9(b)).

Fig. RL9 (Fig. S-13 in the supplementary information): (a) Diffusion correction factor along the longitudinal direction as a function of the NP radius for four different radii of the cylindrical microchannel. The vertical dashed lines refer to the experimentally investigated NP ensembles (average ensemble hydrodynamic diameters: 9.5nm, 14nm, 18nm, 30nm). The case of $R = 8\mu\text{m}$ corresponding to the ARE-fiber used in the experiments presented is emphasized by a thicker line width. (b) Figure 2 of the work of J. M. Nitsche und G. Balgi [Nit1994].

To emphasize this point, example values have been extracted from the Fig. RL9 and are presented in Tab. RL2.

Tab. RL2 (Tab. S-3 in the supplementary information): Selected value of the diffusion factor taken from Fig. RL9(a)

a [nm]	R [μm]	λ	k
9.5	8	0.001	0.992
9.5	0.5	0.019	0.921
30	8	0.004	0.978
30	0.5	0.06	0.817

Clearly visible is that for the ARE-fiber used in this work ($R = 8\mu\text{m}$), the impact of the confinement is small and is included into the analysis. For smaller channels, however, the diffusion coefficient indeed changes, and the resistance factor needs to be considered to obtain the correct diameter. Note that similar results are obtained through full numerical simulations of the diffusion for the transverse case (see Sec. S.11 *Resistance factor correction* of the Supplementary Information of Ref. [Jia2021]).

B. impact of confinement on MSD analysis

Another important point which has to be considered with respect to the confinement is its influence on the data analysis, i.e. the MSD analysis itself. The background is that the MSD values saturate in the case of confined diffusion and thus no linear relationship between MSD and lag times is present, making the application of the Einstein-Stokes equation questionable. Thus, for an accurate diameter determination, it is crucial that the influence of confinement is negligible for the number of lag times considered (here: $N_{lag} = 2$).

To demonstrate this influence, the diffusion of NPs in a circular geometry was simulated according to the parameters used in the experiments, and the corresponding hydrodynamic diameters were calculated. For each channel diameter, 50 simulations were performed, and the results were averaged to improve statistics (Fig. RL10).

Fig. RL10 (Fig. S-14 in the supplementary information): 2D nanoparticle diffusion simulations to reveal the influence of microchannel diameter on MSD analysis and thus on hydrodynamic NP diameters. Each plot shows the dependence between determined hydrodynamic NP and microchannel diameter for the four ensemble NP diameters used in the experiments ((a) 60nm, (b) 36nm, (c) 28nm, (d) 19nm). To resemble the experimental conditions as closely as possible, the parameters of the performed experiments were used in the simulations (frame rate: $\nu = 450 \text{ fps}$, measurement time: $T = 9 \text{ s}$, number of images: $N = 4050$). The increasing viscosity in the vicinity of the silica wall (see Sec. A) was neglected based on the discussion of the previous section, and a constant viscosity of (water: $\eta = 1 \text{ mPa} \cdot \text{s}$) was assumed. For each microchannel diameter, 50 simulations were performed, and the resulting hydrodynamic diameters were averaged. The vertical dashed purple line in each plot indicates the diameter of the used ARE-fiber ($d_c = 17 \mu\text{m}$).

The results clearly show that the channel size has a significant influence on the resulting NP diameter, particular in the case of small channel sizes. To quantify this influence, the results of two selected two microchannel diameters are compared in Tab. RL3.

Tab. RL3 (Tab. S-4 in the supplementary information): Comparison of the hydrodynamic diameters at two selected microchannel and two NP diameters. The numbers are taken from Fig. RL10.

assumed hyd. NP diameter [nm]	determined hyd. diameter @ $d_c=17\mu\text{m}$ [μm]	determined hyd. diameter @ $d_c=0.8\mu\text{m}$ [μm]
60	60.51	78.69
36	36.47	51.81
28	28.49	43.15
19	19.41	32.97

These figures also clearly show that very small microchannel diameters impose very large errors in the resulting hydrodynamic diameter. Ultimately, this effect is a result of an insufficient frame rate ($\nu = 450\text{fps}$), since for too small channel diameters saturation already impacts the values of the MSD at the first two lag times.

However, at the diameters of the ARE used in the experiments ($d_c = 17\mu\text{m}$), this saturation effect practically does not appear and the change in the determined diameter regarding the MSD analysis is negligible. Thus, it can be assumed that due to the sufficient frame rate ($\nu = 450\text{fps}$) in the present experiments, the influence of confinement is not critical for the transverse direction.

C. Summary and actions taken

Overall, it can thus be said that in the presented experiments a sufficiently large channel diameter was deliberately chosen to avoid the influences of the changing viscosity near the wall and the saturation of the MSD values. Thus, the hydrodynamic diameters determined in our work can be considered as to be reliable.

To account for the Reviewer's comment, Secs. A and B have been included into the Supplementary Information and the manuscript has been changed as follows: ... The wall thickness ($w = 730 \pm 20\text{ nm}$) of the ARE element (diameter $d_c = 17 \pm 0.5\ \mu\text{m}$) was chosen such to yield low modal losses (0.4 dB/cm in simulation and experiment, **see Supplementary Information**) at the operation wavelength ($\lambda = 532\text{nm}$). **More details on defining the spectral operating range can be found in this work [Zei2017]. From a waveguide and imaging perspective, the presented fiber design has been optimized with respect to several aspects, including the effect of the ARE-jacket junction, modal losses, or the size of the microchannel (for more details, see the Supplementary Information (for more details, see the Supplementary Information (Secs. SI 13 and SI 14, Figs. S-11 and S-12)). From the diffusion point-of-view, the comparably large channel diameter allows to avoid the influences of a changing viscosity near the wall and the saturation of the MSD values used for the fitting. More details and several example calculations on this can be found in the Supplementary Information (Secs. S-15 and S-16, Figs. S-13 and S-14). ...**

R2.11: The authors also highlight that this is the smallest free-diffusing gold NP even

detected by elastic light scattering. Actually, this description is not acceptable. Many methods have been demonstrated to detect particles with much smaller size.

At this point we would like to mention that perhaps the manuscript does not reflect 100% clearly what we want to say exactly regarding the measurement of NPs with very small diameters. In this work, we have been able to (i) detect very small NPs using pure elastic light scattering without the involvement of other effects (such as interference) and, in particular, to (ii) characterize them with high accuracy. As shown in Fig. 3(g), we achieve a CV of only 15% for extremely small NPs, representing the current state-of-the-art in the field.

Overall, we reach in our work the physical detection limit for this type of NTA implementation, as the experiments are only limited by the intrinsic scattering background of water. To our knowledge, this represents the current world record in terms of characterization of NPs using pure NTA technology.

Note that the Nanosight NS 300 from Malvern [Mal2023], commonly considered as the gold standard in this field of research, uses a substantially lower frame rate (30fps) which in case small NPs considered, is critical, as the NPs can quickly diffuse out of the field of view. In our work, we use (i) a substantially higher frame rate (450fps) and (ii) the confinement via the microchannel, overall allowing to track nanoscale objects accurately over long durations.

As already mentioned, in our response to the first comment of the Reviewer (R2.1), NTA relies on a different operation principle than DLS and allows receiving the properties of individual NPs. DLS, which principally also allows detection of small NPs, analyzes the properties of ensembles, which leads to several limitations as outlined in R2.1. For further details on the comparison of FaNTA, NTA and DLS, we would like to direct the Reviewer's attention to our response to her/his first comment (R2.1).

To reflect the Reviewer's comment appropriately, we have clarified the text in several places and changed it as follows:

Abstract: ... The central result is the characterization of diffusing NPs as small as 9 nm with record-high precision, corresponding to the smallest diameter yet determined for an individual nanoparticle with NTA using elastic light scattering alone. ...

Main text: ... The smallest NP diameter measured in this work is 9 nm, which for gold corresponds to a scattering cross-section of 0.026 nm². This is, to the best of our knowledge, the smallest free-diffusing gold NP detected and individually characterized with record-high precision by NTA that solely uses elastic light scattering. ...

Conclusion: ... The key result represents is the detection and characterization of very small NPs with a diameter as small as 9nm at CV of only 13%, which corresponds to the smallest NP diameter determined so far with NTA based on pure elastic light scattering at a record low CV, reaching the limit of this type of approach. ...

R2.12: I recommend the authors to further improve the quality of this work. There is a big space to polish the results. In the current form, I think it is much suitable for a more

specialized journal with lower quality requirement.

Again, we would like to thank the Reviewer for the valuable feedback provided on our manuscript. We have taken the comments seriously and made significant improvements to the manuscript and supplementary information. Particularly, the latter has been substantially extended by additional considerations, including experiments and simulations. We believe that all the Reviewer's comments have been adequately addressed, and the results have been polished to a higher quality. We sincerely hope that we have been able to improve the manuscript to the Reviewer's satisfaction. Thank you for taking the time and effort to review our work. We are greatly looking forward to your feedback.

Response to comments of Reviewer #3

In the manuscript titled "Characterization of diffusing sub-10 nm nano-objects using single anti-resonant element optical fibers" Wieduwilt and coauthors tested the FaNTA approach for the characterization of very small gold particles, down to the smallest size of 9 nm, with record values for coefficient of variation, approaching the limit of detection of the approach, being limited solely by the scattering of light by the ultrapure water. To achieve this result, they specifically design a fiber that features a single anti-resonant element at one of the distal ends of the hollow channel into which the particles flow. Although the paper has a strong experimental soundness and many aspects that can be defined as "second-order aspects" are well characterized, with only a few experimental "weak points" that can be highlighted, I take the chance to suggest some points that can be improved for better clarity. For this reason, I think the paper can be further considered for publication if the authors can address the concerns listed in the following, in particular concerning differences with their previous works (point 1 below).

We are grateful to the Reviewer for the valuable feedback, especially for recognizing the accomplishments of our work and noting that the work includes only a small number of weak points. In the following, we address all issues raised by the Reviewer in a detailed point-by-point reply. In addition, we would like to note that, triggered by the Reviewer's comment, the motivation of the work and its difference from previous works have now been much better elaborated. Thus we sincerely hope that we have been able to improve the manuscript to the Reviewer's satisfaction.

Apart from this aspect, that I feel should be central in the final decision, firstly, I would like to list the more relevant experimental points, while later in the report I would like to highlight some minor points, mostly related to corrections that can be implemented in the manuscript, figures and other aspects not directly labeled as experimental.

Major points:

R3.1: How far does the proposed paper deviate from the one published by the same authors in August this year? (<https://doi.org/10.1002/sml.202202024>) It seems that the device, the methodology, and the analysis are quite similar between the two papers. Also, the design of the ARE element is quite similar, and in the abovementioned paper more details about the fabrication of the probe and about the data analysis are provided. Although in that case the size of the monitored particles

was an order of magnitude bigger (with the smallest detected particles in this work being 9 nm for gold and the smallest particle detected in the bimodal mixture of the August paper being around 60 nm in the supplementary material, while in the main text 100 nm polystyrene beads are analyzed). The authors should provide sufficient details on the methodological and experimental improvements between the two papers, clarifying also in the main text which are the improvements that justify the publication of the reviewed paper in Nature Communications.

We thank the Reviewer for raising this important point, which prompted us to reconsider the submitted manuscript carefully from this perspective. This has made it clear to us that the differences between the two studies have indeed not been sufficiently presented, for which we would like to apologize.

From our point-of-view, especially after revising the manuscript, the two studies have the following different objectives: In our work published in *Small* [Nis2022], two aspects were mainly investigated, namely (i) the characterization of microscopic imaging through the ARE and the related influences on the tracking and (ii) the potential of the ARE-approach to separate mixtures of nanoparticles (NPs) in terms of hydrodynamic diameter. These results are taken up in the current study in order to show the potential of the ARE-concept with respect to the characterization of NPs with extremely small diameters including the associated influences, where we were able to show the fundamental limit of this approach. In the course of the revision, we added new experimental data and new simulations to the manuscript as requested by the Reviewers, so that now the differences between the study published in *Small* and the added value of the present work are substantially clearer.

In the following, we present the differences of the two studies in detail, listing separately the issues that were already included in the submitted manuscript (Sec. A) and those that were newly added in the course of the revisions (Sec. B). For each aspect mentioned in Sec. B, we have indicated where the corresponding details can be found in the response letter (i.e., in which response to a Reviewers' comment the details can be found).

A. Aspects that are already present in the submitted version and are not discussed in the paper published in Small

- Measurement of a record-low scattering cross-section of a diffusing NP using NTA, reaching the fundamental limit of this approach,
- Realization of a novel chip design to reduce the background signal to the absolute minimum, thus optimizing the signal-noise ratio,
- Uncovering the limits of the presented concept in terms of maximum applicable power and impact of photon pressure on the results, and
- Introduction of z-scoring to filter out outliers in the data analysis to improve statistics (now explained in the supplementary information, FM1.1).

B. Aspects that have been added to the manuscript and/or Supplementary Information in the context of the revision and are not discussed in the paper published in Small

New experimental studies added to the manuscript and/or Supplementary Information

- First-time precise measurements of the loss of a water-filled ARE-fiber using the cut-back method, agreeing very well with simulated data (R2.7, Sec. D),
- Experimental reliability studies showing

- the temporal stability of the transmitted power (R2.5, Sec. B),
- the reliability and precision of the concept by successively measuring the same NP several times (R2.5, Sec. C),
- Discussion of incoupling optimization including an imaging transmitted mode (R3.6), and
- Experimental comparison between FaNTA and DLS in terms of concentration dependence (R2.1).

New simulations added to the manuscript and/or Supplementary Information

- Modal simulations that
 - reveal the susceptibility of the fiber design on silica wall thickness (R2.7, Sec. A),
 - show how the dimensions of the junction between ARE and jacket were chosen (R2.7, Sec. C),
 - reveal the modal birefringence due to the presence of the junction (R2.7, Sec. C),
- In-depth discussion of the impact of the confinement on
 - the diffusion of NP for the ARE-fiber used here (R2.10, Sec. A),
 - the associated mean-square displacement (MSD) data analysis in relation to
 - the impact of saturation (R2.10, Sec. B), and
 - the impact of the number of lag-times considered (R3.3, Sec. B).

To account for the Reviewer's comment, the text has been modified at the following location:

Introduction: ... In this work, we ~~uncover~~ **unlock** the potential of FaNTA for the characterization of very small NPs (<20nm) by means of experimental investigations and detailed statistical analysis (Fig. 1(a)). The study is based on an optimized experimental configuration and a sophisticated fiber design that consists of one single anti-resonant channel, both ~~of are~~ **having** tailored properties to achieve optimized FaNTA performance. **Based on the results of our previous work that concentrated on the microscopic imaging through the ARE and the capabilities to separate bimodal mixtures [Nis2022], we focus here on uncovering the potential of the ARE-concept with respect to the characterization of NPs with extremely small diameters, where we were able to uncover the fundamental limits and influences. In addition to the aspects addressed in [Nis2022] we discuss in this work issues such as the impact of the microchannel confinement on diffusion and data analysis, the realization of a novel chip design, or the influence of photon pressure, and present an experimental reliability study of the ARE-concept. All the issues addressed in this study are related to the characterization of very small NPs.** The key result is the characterization of diffusing NPs as small as 4 as 9 nm with record low CV values (CV = 13%, **an example of tracking such small NPs with FaNTA is shown in Figs. 1 (b) to (d)**). The detectable scattering cross-section is here only limited by the background scattering of the ultrapure water, thus reaching the fundamental limit ~~in the context~~ of this approach. ...

R3.2: On page 10 the authors say “Note that with increasing demand on precision, fewer trajectories have the necessary length $N_{f,min}$, leading to smaller number of NPs to be analyzed”. It is not clear if this point is a limitation or a point of strength of the proposed method. This opens to the other question on the statistical analysis, reported also in figure 3.b-e:

We thank the Reviewer for mentioning this aspect, which from our perspective is an overall strength of the FaNTA approach, since other NTA implementations only provide much shorter trajectories and thus a study associated with Fig. 3 cannot be performed straightforwardly. With our approach, however, it is actually possible to choose whether a large number of NPs (i.e., trajectories) with moderate accuracy or a small number of NPs with very high accuracy should be analyzed. Note that the possibility to acquire long trajectories additionally offers the possibility to analyze dynamic processes at the individual NP level, representing a type of study that is difficult to conduct with other NTA-concepts.

In the light of this, we would like to mention the following aspect, which is key to FaNTA: Generally, NTA gives a precision for the ensemble of nano-objects (mainly resulting from the low number of frames), while FaNTA gives a precision per nano-object. The same argument holds for DLS, emphasizing that FaNTA represents a new type of analytical platform with unique properties and opportunities. For a more in-depth discussion of the differences and advantages of FaNTA, we would like to direct the Reviewer’s attention to your response to the first comment of Reviewer #2 (R2.1), which has raised a question along the same lines. In the following, we will discuss the individual comments of the Reviewer in detail.

R3.2a: Is the number of samples enough to be statistically fitted with the curves used to extract d_p and σ_d ? For Figure 3.d only 11 particles are detected with $N_{f,min}$ of 1000. While the central value of the distribution, which determines d_p , is almost the same for all the figures, the value of σ_d seems dependent on the number of particles detected. The fewer the detected particles, the narrower the histogram is, the smaller the value of σ_d . And if the histograms in figures 3.d and 3.e follow evidently the fitting curve, the same cannot be said for figure 3.b.

We thank the Reviewer for this important comment, which we answer in detail below.

A. How to get the curves

We would like to mention that the curves shown in Figs. 3(b)-(e) are actually not fitted to the histograms. Instead, the diameter and CV values were calculated directly from the averaging process of the data and the corresponding curves were plotted in the associated histogram. In this process, the trajectories are weighted according to their length, with the consequence that longer trajectories, i.e., trajectories with higher accuracy, have a larger contribution. Thus, by increasingly excluding inaccurate samples in the averaging (via increasing $N_{f,min}$), the accuracy of the determination process for the diameter is increased, which as a consequence decreases the associated standard deviation (Fig. 3(f)). This process is valid as long as the mean diameter does not change, which is the case in the data presented in this work.

B. Influence of the trajectory length on the accuracy of ensemble statistics

Generally, the same number of NPs is always considered in the analysis, while by increasing $N_{f,min}$, trajectories with low accuracy are increasingly omitted. As shown in [Nis2022], the standard deviation of the ensemble measurement σ_m is given by

$$\frac{\sigma_m}{\bar{d}_m} = CV_m = \sqrt{CV_E^2 + CV_S^2} \geq \sqrt{CV_E^2 + \frac{2}{N_{f,min} - 1} \delta}$$

with the corresponding average diameter \bar{d}_m , the measured CV CV_m and the CVs of the NP ensemble and the system CV_E and CV_S . Due to the high localization accuracy of the FaNTA approach, it can be roughly assumed that $\delta \approx 1$. If the mean NP diameter is constant for all $N_{f,min}$ considered (as it holds in the experiments discussed here), the standard deviation of the measurement σ_m depends on $N_{f,min}$ according to the above equation. In the case that the CV value of the system is exclusively correlated to data analysis (i.e., the instrumental measurement error is negligibly small), $CV_S^2 = \frac{2}{N_{f,min} - 1}$.

For this situation, Fig. RL11 shows that longer trajectories are essential to approach the CV-values of the NP ensemble, i.e., precise ensemble characterization requires long trajectories.

Fig. RL11 (Fig. S-15 in the supplementary information): Standard deviation of the measurement as a function of the minimal number of frames for three different CV-values of the NP ensemble (defined in the legend).

This trend is also evident in Fig. 3(f), where the relationship between $N_{f,min}$ and the CV value of the measurement is shown for the experimental data. It should be mentioned again that the dependency shown in Fig. RL11 is only meaningful as long as the mean diameter is almost the same for all $N_{f,min}$ values considered, which holds for the present experiments.

C. Actions taken

Although the relationship between the CV value and the minimum trajectory length is already shown in Fig. 3(f), we decided to include the discussion of Sec. B in the Supplementary Information (Sec. SI 17). The main text has been extended accordingly: ... To exemplify this dependency, Figs. 3(b)-(e) show the result of the MSD analysis for a selected ensemble (20 nm gold) taking into account only NP with trajectory length above a certain threshold $N_{f,min}$ (indicated by the colored horizontal

lines in Fig. 3(a)). It is evident from this example that the precision of the retrieved NP diameter (σ_d) strongly depends on the required trajectory length $N_{f,min}$ (e.g., $\sigma_d = 5.9nm @ N_{f,min} = 50$ (Fig. 3(e)), $\sigma_d = 2.0nm @ N_{f,min} = 1000$ (Fig. 3(b))). **Note that the curves presented in the histograms do not result from fitting, but rather from averaging of the data. In this process, the trajectories are weighted according to their length, so that longer trajectories, i.e. trajectories with higher accuracy, have a stronger contribution.** This is where the great advantage of fibers becomes clear, as unlike other NTA approaches, FaNTA allows very long trajectories to be retrieved per NP through the channel-mediated confinement, resulting in statistics with excellent significance (**more details on the correlation between trajectory length and the precision of the ensemble statistics can be found in the Supplementary Information (Sec. SI 17, Fig. S-15)). ...**

R3.2b: Strictly related to point 6.a, how this affects the results in case of a blind experiment in which the mean size of the particle is not already known, as in the case of the experiments reported in the manuscript?

Considering the argumentation of the previous section, we believe that the number of trajectories used is not critical as long as the determined NP diameter does not change significantly for fewer trajectories. For the details, we would like to refer the Reviewer to our response to the Reviewer's previous question (R3.2a), which addresses this point in great detail.

Overall, based on this detailed discussion and the additions to the manuscript and Supplementary Information conducted in the context of R3.2a, we hope that we have been able to adequately address the Reviewer's questions.

R3.2c: Figures 3.d and 3.e, and partially also figure 3.c show a sort of dip for the bin in between 26 and 27 nm. Could this be related to a bimodal distribution, very close in size, of the analyzed particles?

This is an interesting observation that we did not notice when preparing the manuscript. In principle, it could be true that two mean diameters exist, although such a bimodal distribution was not specified by the manufacturer of the NPs. To clarify this aspect, further experiments would be necessary at this point to investigate just this particular sample. Note that the width of the bins within the histograms also plays a role, and the histograms would look different with choosing a different bin width.

Overall, since such a study is not directly related to the discussion conducted in this manuscript, we would like to refrain from any speculation at this point and thus have not included this aspect in the manuscript. We sincerely hope that the Reviewer agrees with our reasoning.

R3.3: On page 11, the authors state “an earlier saturation is observed for the smaller NPs due to their faster diffusion”, but this observation cannot be directly extracted from figure 4 (in particular 4.a), where the curve for 30 nm saturates (or better, suggests the saturation for higher values of lag frames) before 20 nm and the curves for 20 nm and 50 nm show a similar trend. Probably, the x-axis limits can be extended for appreciating more the trend of the curves.

We thank the Reviewer for this comment. We would like to mention that extending the lag-time range in order to demonstrate the saturation of the MSD values for the

experimental data is critical because the curve shows strong fluctuations above a certain lag-time limit. This effect is intrinsic to the MSD analysis and results from the fact that as lag-times increase, fewer and fewer data points contribute to the MSD value of interest, making them statistically less significant. Thus, the data points with the smallest lag-times always have the highest accuracy and are generally used for determining hydrodynamic diameters via linear fitting. Before we go into the specific response to the Reviewer's comment, we would like to direct the Reviewer attention to the following aspect.

A. Influence of confinement (response to the tenth comment (R2.10) from Reviewer #2 (R2.10)): We would like to mention here that the diameter of the ARE was intentionally chosen to be large enough ($D_{ARE} = 10\mu m$) that the influence of confinement on the data analysis can be neglected for the NP diameters used here. Two aspects need to be considered in the context of confinement: (i) On the one hand, the ratio between NP and channel diameter is sufficiently small ($\lambda = a/R < 0.02$, (a : radius of NP, R : radius of channel)) that a modification of the viscosity in the vicinity of the wall, which is typically summarized in the resistance factor R_s , does not need to be taken into account (see Sec. A of R2.10 and Fig. RL9). (ii) In addition, the selected frame rate ($\nu = 450 \cdot 1/s$) is high enough that the confinement effect does not appear for the first lag times which is essential for the linear fitting (see Sec. B of R2.10 and Fig. RL10). To avoid unnecessarily prolonging the response letter, we would like to refer the Reviewer to our response to the tenth comment of Reviewer #2 (R2.10, Sec. A and Sec. B) and the amendments in the Supplementary Information (see Secs. SI 15 and SI 16, Figs. S-13 and S-14).

B. Simulation of particle diffusion for large lag times

The experiments presented in this study do not reach complete saturation within the given measurement time due to the large channel diameter. However, it is apparent that the curve deviates from its initial linear evolution towards higher lag times. To show the saturation effect, we decided here to simulate the 2D diffusion of the NPs used in the experiments for a much longer acquisition time ($T_{total} = 120s$) using two microchannel diameters (Fig. RL12, $d_c = 4\mu m$ and $d_c = 17\mu m$). Note that the larger channel diameter corresponds to the value of the ARE fiber used in the experiments. For the small channel diameter, the saturation of the MSD curve can be clearly seen, with, as mentioned in the manuscript, this effect starting earlier for smaller NPs due to faster diffusion. For the larger channel, this effect is much less pronounced and essentially a deviation from the linear evolution of the MSD-lag time dependence is observed.

Fig. RL12 (Fig. S-16 in the supplementary information): Mean square displacement (MSD) analysis of simulated data of a NP diffusing in a circular geometry for four different NPs. The two plots refer to two diameters of the disk ((a) $d_c = 4\mu\text{m}$, (b) $d_c = 17\mu\text{m}$). The analysis is performed along a selected transverse axis (here x -axis). The parameters are as follows: frame rate: $\nu = 450\text{fps}$, measurement time: $T_{\text{total}} = 120\text{s}$, number of images: $N = 60000$). The increasing viscosity near the silica wall was neglected, and a constant viscosity of water ($\eta = 1\text{mPa}\cdot\text{s}$) was assumed.

C. Actions taken

To address the Reviewer's comment, the above discussion (Sec. B) has been included in the Supplemental Information (Sec. SI 18, Fig. S-16) and the following paragraph of the main text in the manuscript has been modified as follows: ... While longitudinal and transversal MSD values are similar on short timespans (Fig. 4 (b)), the transversal MSD ~~saturates~~ **deviates from the linear behavior** for longer lag times, an effect that is associated with the finite extension of the microchannel restricting the free diffusion (Fig. 4(a)). As expected, an earlier ~~saturation~~ **onset of the deviation** is observed for the smaller NPs due to their faster diffusion. **To show this effect more clearly, corresponding simulations are presented in the Supplementary Information (Sec. SI 18, Fig. S-16).** ...

R3.4: On page 15, the authors wrote “Thus, signal and background have similar intensity. We verified this by using ethanol instead of water, leading to a higher background signal proportional to its higher scattering coefficient b_{ethanol} ”. It is not clear what has been verified. Did the authors verify only that ethanol produces a higher background signal? Or did they perform a measurement of NPs dispersed in ethanol instead of water? Which is the detection limit in this latter case?

We thank the Reviewer for the comment on the ethanol measurements. In the following, we first explain the reason for presenting the ethanol-related measurements and then present an estimate for the smallest detectable NP diameter in the situation ethanol is the host medium.

A. purpose of the Ethanol measurements

We would first like to mention that the purpose of the presented comparison of the two liquids is solely to show that the observed background scattering signal results from

the liquid medium, i.e. from water, and not from light, which dissipates transversely during leaky mode propagation. As presented in the Supplementary Information (Fig. SI3, Fig. S-2), we were indeed able to verify that, in the case of ethanol, a stronger scattering originating from the core domain is observed. This result is consistent with values for Rayleigh scattering reported in the literature, which predict a threefold higher scattering signal for ethanol compared to water (Supplementary Information Sec. SI3). This result unambiguously reveals that the smallest possible NP diameter that can be determined with the existing system is determined by the scattering background of the liquid medium, i.e., water, thus reaching the fundamental limit of the elastic light scattering-based NTA.

B. Estimation of smallest detectable NP diameter in the case of Ethanol

The following argumentation can be considered to estimate the smallest possible NP diameter in the case ethanol is the background medium:

According to the argumentation in the main text, the following equation holds

$$b \cdot L_z \approx \frac{\sigma_{scat}}{A_{ch}} = K \frac{d_p^6}{A_{ch}}$$

with the scattering coefficient b , the longitudinal pixel length L_z , the channel area A_{ch} , the scattering cross-section σ_{scat} and the diameter of the NP d_p . By setting up one such equation for each of the two liquids (K is a constant being identical for both liquids), the following relationship is obtained

$$d_{p,ethanol} = \sqrt[6]{b_{Ethanol}/b_{H2O}} \cdot d_{p,H2O}.$$

The ratio under the square root can be estimated from the Rayleigh ratios given in the Supplementary Information (Sec. SI3). The result NP diameter is $d_{p,ethanol} \approx 10.8nm$ in case that $d_{d,H2O} = 9nm$ is assumed.

C. Summary and actions taken

We would like to mention here again that the purpose of comparing the two liquids is only to elucidate the origin of the observed background scattering signal. Thus, we refrain from discussing the smallest possible observable NP diameter in the case of ethanol in the manuscript and hope that we could sufficiently explain the related reasons to the Reviewer.

Minor points:

R3.5: In the introduction, the authors state that “The study is based on an optimized experimental configuration and a sophisticated fiber design that consists of one single anti-resonant channel, both of are tailored properties to achieve optimized FaNTA performance” but no details on the optimization, nor a comparison between different types of AREs are provided. How are the characteristics of the ARE chosen? It seems that they are obtained to achieve low modal losses at 532 nm, following Ref. 24, but it could be useful to add those experimental results and simulations to the supplementary material.

We thank the Reviewer for the comment, which was also raised by Reviewer #2 in her/his seventh comment (R2.7). To address it, we have included the relevant simulations and measurements into the manuscript. Moreover, as requested by the Reviewer, we have substantially extended the Supplementary Information in order to

make clear why we have chosen and implemented this particular fiber structure in the context of FaNTA. In order not to unnecessarily prolong the response letter, we would like to direct the Reviewer's attention to our response to the seventh question of Reviewer #2 (R2.7), which addresses exactly the aspects mentioned by the Reviewer.

R3.6: Is the optimization routine of the butt coupling between the launching system and the fiber channel manual or is it somehow automatized?

In the following, we would like to describe the procedure how the coupling of light into the ARE was optimized: The optimization of the butt coupling is performed manually and includes the adjustment of the delivery fiber in xyz -direction and the rotation of the SEF around its axis. Here, the delivery fiber is mounted on a fiber launching stage (MDE122, Elliot Scientific), ensuring a sufficient adjustment distance of 2 mm/axis with a resolution of 20 nm. The SEF is located on a holder which allows manual rotation of the fiber, aiming to position the SEF in such a way that the contact point of ARE and jacket faces the objective. The light at the output of the ARE, i.e., SEF is used as an indicator for the quality of the alignment regarding exciting the fundamental ARE-mode and is imaged onto a camera chip using a lens. An example image of a measured profile of the fundamental mode is shown in Fig. RL13.

Fig. RL13 (Fig. S-17 in the supplementary information): Measured intensity distribution of the fundamental mode inside the ARE along the symmetry axis of the cross-section, taken during butt coupling optimization. The inset shows the image acquired, with the dashed line indicating the axis along which the line scan is performed.

To address the Reviewer's comment, the above text was included into the Supplementary Information (Sec. SI 19, Fig. S-17) and the main text in the manuscript was changed as follows: ... To optimize the butt coupling efficiency ($\sim 90\%$) with respect to launching efficiency of the fundamental mode, the output mode (example mode image shown in Fig. 2(c)) is imaged by another camera (DCC1240C-HQ; Thorlabs). **More details on the optimization of the incoupling can be found in the Supplementary Information (Sec. 19, Fig. S-17).** The required guided laser power at the position of the particles ranges from 0.2 mW for the 50 nm gold NP to 70 mW for the 9 nm gold NP. ...

R3.7: Are the illumination laser powers predicted analytically for each NP size or are they estimated empirically? If the first is true, which is the analytical relation between P_{\min} and d ?

In the experiments discussed here, the excitation power has been determined empirically. Due to the pronounced dependence of the scattering intensity on the diameter of the NPs (R^6 -dependence), a significantly higher excitation power is needed for small particles than for large particles. Furthermore, the specification of a minimum power is generally not feasible as it depends on various experimental circumstances such as the sensitivity of the camera system.

To account for this point, the text in the manuscript has been modified as follows: ...
The required guided laser power at the position of the particles ranges from 0.2 mW for the 50 nm gold NP to 70 mW for the 9 nm gold NP, **which was empirically determined.** ...

R3.8: Minor: the authors say “Note, that the time a particle can be tracked, can be tuned by the NP concentration”. Could the authors elaborate on this point?

We thank the Reviewer for the comment to discuss this point in more detail. The sentence “*Note, that the time a particle can be tracked, can be tuned by the NP concentration*” says that in the case of a very high NP concentration, NP can cross each other and the tracking code may not be able to differentiate which NP belongs to which trajectory. If this occurs, the corresponding trajectory is stopped and a new trajectory is started. This effect typically occurs at very high concentrations beyond those used in the experiments discussed in the manuscript.

To formulate this statement more precisely, the above sentence of the manuscript has been modified as follows: ... ~~Note, that the time a particle can be tracked, can be tuned by the NP concentration.~~ **at concentrations beyond those used here, NPs may cross each other many times and the tracking code may not be able to differentiate which NP belongs to which trajectory, resulting in shorter trajectories.** ...

R3.9: On page 9 the authors state “Applying the above analysis to all measured NP ensembles confirms, that the CV improves with increasing $N_{f,\min}$ ”, while Figure 3.f is somehow misleading, being shown in such a way that lets the reader understand that CV is the independent variable, while it depends from N_f .

We thank the Reviewer for this comment and agree that the abscissa in Fig. 3(f) should be the minimum number of frames $N_{f,\min}$. We have changed Fig. 3f accordingly and also adjusted the figure caption. Please also note that we have changed the colors of the points in Figs. 3(f) and (g) to clearly distinguish them from the plots Figs. 3(b) - (e).

Fig. 3 of the main text: (a) Example of a measurement of an ensemble of 20 nm NP, showing trajectory length N_f as a function of retrieved diameter. The differently colored lines indicate the considered minimum trajectory length $N_{f,min}$ used to calculate the histograms (probability of hydrodynamic diameter) shown in (b) to (e). (f) CV as a function of $N_{f,min}$. (g) Number of particles per frame as a function of CV. It is evident from the plots that low CV values require long trajectories, which, however, reduces the number of successfully evaluated particles in each frame.

R3.10: In the introduction, the acronym “NPs” is used in different contexts (in line 3 for Nanoparticles, in line 4 for water contaminants, in line 11, generically to identify nanometric objects). If it is not intended, please use different acronyms for different concepts.

We thank the Reviewer for carefully reading our manuscript. In the context of this work, the acronym "NPs" should generally refer only to nanoparticles such as gold nanospheres.

Therefore, to address the aspect of the Reviewer, we have included the following changes in the main body of the manuscript:

... monitoring of water contaminants (~~NPs~~) ...

... One approach applied in this context is dynamic light scattering (DLS), evaluating the diffusive properties of ensembles of NPs. Here, even though **nano-objects** NPs such as proteins or molecules can be characterized ...

... In addition to the characterization of ensembles of **nano-objects** NPs, FaNTA uniquely ...

R3.11: References to figure 1.b, 1.c, and 1.d cannot be found in the main text. Please cite them in the main text or move it to the supplementary material (I would suggest

citing them in the main text).

We thank the Reviewer for pointing out this flaw, which we have corrected with the following text changes: ... In this work, we ~~uncover~~ **unlock** the potential of FaNTA for the characterization of very small NPs (<20nm) by means of experimental investigations and detailed statistical analysis (Fig. 1(a)). The study is based on an optimized experimental configuration and a sophisticated fiber design that consists of one single anti-resonant channel, both ~~of-are~~ **having** tailored properties to achieve optimized FaNTA performance. **Based on the results of our previous work that concentrated on the microscopic imaging through the ARE and the capabilities to separate bimodal mixtures [Nis2022], we focus here on uncovering the potential of the ARE-concept with respect to the characterization of NPs with extremely small diameters, where we were able to uncover the fundamental limits and influences. In addition to the aspects addressed in [Nis2022] we discuss in this work issues such as the impact of the microchannel confinement on diffusion and data analysis, the realization of a novel chip design, or the influence of photon pressure, and present an experimental reliability study of the ARE-concept. All the issues addressed in this study are related to the characterization of very small NPs.** The key result is the characterization of diffusing NPs as small as 9 nm with record low CV values (CV = 13%, **an example of tracking such small NPs with FaNTA is shown in Figs. 1 (b) to (d)**). The detectable scattering cross-section is here only limited by the background scattering of the ultrapure water, thus reaching the fundamental limit ~~in the context~~ of this approach. ...

R3.12: In figure 1, no caption is written for figure 1.d.

We thank the Reviewer for pointing out the missing text in the figure caption, which we have supplemented as follows: ... Fig. 1: (a) Sketch of the concept single antiresonant-element (ARE) fiber-assisted nanoparticle tracking analysis (FaNTA) applied here to track sub-10nm nano-objects. **An example of tracking such small nanoparticles with FaNTA can be seen in the images to the right.** (b) **Selected** frame showing 9nm gold NPs diffusing inside the anti-resonant element. (c) Processed image showing the localization of the NPs (red circles). **(d) Corresponding measured trajectory of several tracked nanoparticles. In all images on the right-handed side, the horizontal yellow dashed lines indicate the wall of the ARE.** ...

R3.13: Although cited in the text, no labels for the characteristics of the ARE are provided in the SEM micrograph in figure 2. Please add labels and size for w and d_c in figure 2.

We thank the Reviewer for this valuable comment, which has triggered us to revise the entire figure. To specifically address the comment of the Reviewer, we have introduced a close-up view of the ARE containing the labels mentioned. Furthermore, the citations of the figures within the main text have been adjusted accordingly. Below you can find the new figure and the changed text.

Fig. 2 of the main text: (a) Experimental setup used in the FaNTA experiments discussed here. (b) SEM image of the used ARE-fiber (**the scale bar shown in (b) refers to 20 μ m and also applies to (b) and (d)**). (c) is a close-up view of the ARE. (d) Measured mode profile at the operation wavelength. The bottom row shows example raw images of tracked gold nanospheres ((e) 9 nm, (f) 20 nm, (g) 30 nm). Tracked particles are marked red. The orange circle shows a particle with typical SNR and brightness which **increases** with growing NP diameter. The yellow dashed line marks the channel wall.

The following changes have been made to the text:

... providing effective single-mode guidance directly inside the water-filled ARE (Fig. 2(d), details can be found in ...

... the output mode (example mode image shown in Fig. 2(d)) is imaged by ...

... The potential of our approach is clearly visible from the three example frames shown in Figs. 2(e)-(g): NPs with very small physical diameters can be tracked by our approach, with Fig. 2(e) showing ...

... We successfully imaged the 9 nm gold NP at 450 fps (Fig. 2(e)) and used this frame rate ...

... (a scanning-electron-micrograph (SEM) images are shown in Figs. 2(b) and (c)) and the microchannel has an NA of 0.024 (water-filled). ...

R3.14: The final sentence in the first paragraph of the methods section should read “which is substantially higher than THE ONE used in the NTA experiments.”

Thank you, the sentence has been corrected as follows: ... For comparison, DLS measurements using a commercial device have been performed at the recommended concentration of 10⁵NP/nL (Zetasizer Nano ZS and Ultra; Malvern Pananalytical) which is substantially higher than **the one** used in the NTA experiments. ...

R3.15: For figure 3, panel a is cited in the text before panel b.

The text has been changed as follows: ... To exemplify this dependency, **Figs. 3(a)-(e)** show the result of the MSD analysis for a selected ensemble (20 nm gold) taking into account only NP with trajectory length above a certain threshold $N_{f,\min}$ (indicated by the colored horizontal lines in Fig. 3(a)). ...

R3.16: Many references to values that should be in table 1 and in the supplementary information tables are not in there. Examples: dh, dp.

We thank the Reviewer for this point. We have reviewed the entire manuscript again and added the missing parameters to the table in the Supplementary Information (Sec. SI 21). The format of the table has been additionally improved.

R3.17: The acronym TEM for transmission electron microscope is not defined.

We thank the Reviewer for spotting this. The text has been changed as follows: ... always larger than its physical counterpart d_p (**measured by a transmission electron microscope (TEM)**), ...

R3.18: On page 14, “nanoschannels” should read “nanochannels”.

Thank you. The revised text reads as follows: ... which can be an issue in fibers with **nanochannels** ...

R3.19: In supplementary SI 3, specifically figure S2, I would suggest adding something that can help the reader to have a direct comparison between the background level of water and ethanol. A colorbar, or a small indication of the mean value of the pixels inside the channel could work.

We thank the Reviewer for the comment regarding the comparison of the scattering intensities of both situations. To respond to the comment adequately, new measurements were performed to compare the scattering intensities quantitatively. For this purpose, both cases were measured again under exactly the same experimental conditions and the intensity counts in the area of the microchannel were summed up (yellow rectangle in Fig. RL14, accumulated counts are given on the left-handed side). The ratio of the resulting values is 2.9, which agrees well with the ratio of the Rayleigh ratios reported in literature (Rayleigh ratios: $R_{Ethanol} = 3.92 \cdot 10^6 \text{ cm}^{-1}$, $R_{water} = 1.34 \cdot 10^6 \text{ cm}^{-1}$, $R_{Ethanol}/R_{water} \approx 2.92$ [Par1968]).

Fig. RL14 (Fig. S-2 in the supplementary information): Image of the scattered background light acquired by filling the ARE fiber with (a) water and (b) ethanol under identical conditions. No nanoparticles are involved in this experiment. The yellow rectangle indicates the region of the ARE in which the intensity counts were summed up (the respective value is given on the left-handed side of the images).

To account for the Reviewer's comment, the figure in the Supplementary Information was updated by the new experiments and the text was changed according to the above discussion: ... Fig. S-2 shows the background signal of the ARE inside the SEF, by filling the fiber with ultra-pure water (a) and ethanol (b) only (NP-free liquid). **As the experimental conditions were the same in both cases, a quantitative comparison of the amount of scattered light is possible by summing up the intensity counts in the area of the microchannel (yellow rectangular in Fig. S-2, accumulated counts are given on the left-handed side). The ratio of the resulting values is 2.9, which agrees well with the ratio of the Rayleigh ratios reported in literature (Rayleigh ratios: $R_{Ethanol} = 3.92 \cdot 10^6 cm^{-1}$, $R_{water} = 1.34 \cdot 10^6 cm^{-1}$, $R_{Ethanol}/R_{water} \approx 2.92$ [Par1968]), proving that the background signal truly originates from the liquid and not from fiber losses or experimental inaccuracies. ...**

Further modifications to the manuscript

FM1.1: We have also included a description of the z-scoring procedure in the Supplementary Information, as this procedure represents a significant advance in data analysis which was not included in any of our previous papers and is essential for the results shown in this work. We have therefore added a section (Sec. SI 20) to the Supplementary Information, in which the essentials of z-score filtering are described in detail. In particular, we presented a direct comparison of filtered and unfiltered data (Fig. RL15), showing that statistical outliers can be effectively suppressed by this filtering procedure.

Fig. RL15 (Fig. S-18 in the supplementary information): Comparison of results of MSD data analysis (a) with and (b) without z-score filtering of the same experimentally determined data set (physical diameter of gold nanoparticle $d_p = 50nm$).

The main text has been changed as follows: ... Outliers, coming from contaminations or particle agglomerations, are removed by a z-score filtering [REF], which operates through the assumption that the determined diffusion coefficients ought to be normally distributed [REF] (more details on the z-score filtering can be found in the Supplementary Information (Sec. SI 20)). ...

References

Nis2022	M. Nissen, R. Förster, T. Wieduwilt, A. Lorenz, S. Jiang, W. Hauswald, M. A. Schmidt. Nanoparticle Tracking in Single-Antiresonant-Element Fiber for High-Precision Size Distribution Analysis of Mono- and Polydisperse Samples . Small 2202024 (2022) [link]
Ye2012	W. M. Ye, M. Wan, B. Chen, Y. G. Chen, Y. J. Cui, J. Wang. Temperature effects on the swelling pressure and saturated hydraulic conductivity of the compacted GMZ01 bentonite . Environmental Earth Sciences 68, 281 (2013) [link]
Kim2022	J. Kim, R. Foerster, T. Wieduwilt, B. Jang, J. Buerger, J. Gargiulo, L. de Souza Menezes, C. Rossner, A. Fery, S. A. Maier, M. A. Schmidt. Locally structured on-chip optofluidic hollow-core light cages for single nanoparticle tracking . ACS Sensors 7, 2951 (2022) [link]
Gui2021	F. Gui, S. Jiang, R. Förster, M. Plidschun, S. Weidlich, J. Zhao, M. A. Schmidt. Ultralong tracking of fast diffusing nano-objects inside nano-fluidic channel enhanced microstructured optical fiber . Adv. Photon. Res. 2100032 (2021) [link]
Fae2015	S. Faez, Y. Lahini, S. Weidlich, R. F. Garmann, K. Wondraczek, M. Zeisberger, M. A. Schmidt, M. Orrit, V. N. Manoharan. Fast, label-free tracking of single viruses and weakly scattering nanoparticles in a nano-fluidic optical fiber . ACS Nano 9, 12349 (2015) [link]
Zeis2017	M. Zeisberger, M. A. Schmidt. Analytic model for the complex effective index of the leaky modes of tube-type anti-resonant hollow core fibers . Sci. Rep. 7, 11761 (2017) [link]
Jia2020	S. Jiang, J. Zhao, R. Förster, S. Weidlich, M. Plidschun, J. Kobelke, R. Fatobene Ando, M. A. Schmidt. Three-dimensional spatiotemporal nano-scale position retrieval of confined diffusion of nano-objects inside optofluidic microstructured fibers . Nanoscale 12, 3146 (2020) [link]
Bre1977	H. Brenner, L. J. Gaydos. The constrained brownian movement of spherical particles in cylindrical pores of comparable radius: Models of the diffusive and convective transport of solute molecules in membranes and porous media . Journal of Colloid and Interface Science 58, 312 (1977) [link]
Nit1994	J. M. Nitsche and G. Balgi. Hindered Brownian Diffusion of Spherical Solutes within Circular Cylindrical Pores . Ind. Eng. Chem. Res. 33, 2242 (1994) [link]
Jia2021	S. Jiang, R. Förster, A. Lorenz, M. A. Schmidt. Three-dimensional tracking of nanoparticles by dual-color position retrieval in double-core

	microstructured optical fiber . Lab on a chip, 21 , 4437 (2021) [link]
Mal2023	NanoSight NS300, online data sheet [link]
Par1968	G. D. Parfitt, J. A. Wood Light scattering from binary mixtures of water, methanol and ethanol . Trans. Faraday Soc. 64 , 2081 (1968) [link]

Reviewer #2 (Remarks to the Author):

All of the questions have been addressed. No further comment.

Reviewer #3 (Remarks to the Author):

I would like to thank the authors for the thoroughly detailed report they provided after the reviewers' report. Considering the high quality of the motivations provided and the precision in addressing the reviewers' comments, I would definitely suggest the revised paper for publication in Nature Communications.